# 2040 greenhouse gas reduction targets and energy transitions in line with the EU Green Deal

Renato Rodrigues [1] ✉, Robert Pietzcker[1], Joanna Sitarz [1,2], Anne Merfort [1,2], Robin Hasse [1,2], Johanna Hoppe [1,2], Michaja Pehl [1], Ahmad Murtaza Ershad[1], Jarusch Muessel [1,2], Felix Schreyer[1,2], Lavinia Baumstark [1] & Gunnar Luderer [1,2]

The European Green Deal aims to guide the European Union towards achieving net-zero greenhouse gas emissions by implementing a comprehensive set of policy initiatives and legislation. While emission reduction targets and policies up to 2030 are mostly implemented, it is of high priority for EU legislation to spell out the further transformation to climate neutrality by defining interim policy targets for 2040. To provide information for this target-setting process, we use an integrated energy-economy-climate model with high sector detail to explore pathways to achieve climate neutrality in the EU under uncertainty about key energy system developments. Results suggest that emission reductions of 86% (sensitivity range: 80% to 93%) by 2040 relative to 1990 are consistent with a cost-efficient distribution of mitigation efforts over time, substantially exceeding the 78%-level implied by a linear interpolation between the 2030 and 2050 targets. Additionally, we identify a 7-fold (sensitivity range: 4–8-fold) upscaling of electricity generation from wind and solar, a 49% (sensitivity range: 45–59%) share of electricity in final energy supply and an upscaling of carbon capture and storage (CCS) to 188 Mt CO2/yr (sensitivity range: 56–257) as crucial transformation milestones for 2040.

The European Union (EU) is distinguished by its ambitious climate mitigation targets, positioning itself as a leader in global climate action. As a region that constitutes nearly one-sixth of the global economy, the EU's decarbonization strategy is poised to exert significant global influence. The EU's commitments and policies are expected to have far-reaching effects, inspiring and influencing the climate ambitions and actions of other regions worldwide.

With the presentation of the European Green Deal[1] in late 2019, the EU committed to making Europe the first climate-neutral economy at continent-scale by 2050. Since then, the EU has embarked on a legislative journey, spelling out intermediary targets and policy measures to reach its goal. Intermediate climate targets are considered essential as they ensure that mitigation action is taken in a timely manner across the different sectors, thereby preventing lock-ins, stranded assets, and delays that could endanger the achievement of climate neutrality by mid-century[2].

The first significant milestone focused on setting targets for 2030: The European Climate Law[3] not only enshrined the overarching goal of climate neutrality by 2050 but also established the intermediate target of reducing net greenhouse gas (GHG) emissions by at least 55% by 2030, compared to 1990 levels. Simultaneously, the Fit for 55 package[4] presented a set of legislative proposals laying down concrete measures to align all sectors with the 2030 target. These include, among others, the extension of the EU Emissions Trading System to new sectors and

[1]Potsdam Institute for Climate Impact Research, Potsdam, Germany. [2]Global Energy Systems Analysis, Technische Universität Berlin, Berlin, Germany. ✉e-mail: renato.rodrigues@pik-potsdam.de

the tightening of its caps, the increase of renewable energy sources in the overall energy mix, the increase of energy efficiency targets, as well as the introduction of emissions reduction targets for cars and vans. In 2022, following Russia's invasion of Ukraine, the EU additionally launched the RePowerEU plan[5], aiming at rapidly reducing Europe's dependence on Russian fossil fuels. RePowerEU foresees even higher 2030 targets for renewables and energy efficiency than previously proposed by the Fit for 55 package.

The European Climate Law mandated the establishment of an intermediary climate target for 2040, whose provisional agreement was reached by the end of 2025[3,6]. This target-setting process has been substantially informed by the European Scientific Advisory Board on Climate Change[7], which has drawn heavily from preliminary results of the scenarios developed in this paper. As was the case with the EU 2030 targets, science-based analysis is an important enabler of informed decision-making in the process of setting and revising climate targets and policies. The current study aims to contribute to this process by providing an independent counterpoint to the European Commission's impact assessment on the 2040 climate target[8], extending the range of evaluated sensitivities to support robust policy decisions.

Setting an aggregated EU-wide emission reduction target is an important, but not sufficient, step to achieve climate neutrality by 2050. The proposed 2040 emission target will likely be accompanied by sectoral transformation milestones, underpinned by policies and measures similar to those defined for 2030 as part of the Fit-for-55 package. Defining these targets requires detailed quantitative modelling of the transformation across all sectors, considering also relevant uncertainties.

While a few pioneering studies have explored EU deep mitigation pathways[9–11], their results pre-date the establishment of the European Climate Law and its associated policies. One recent model comparison study includes Green Deal targets for 2030 and 2050[12], but the study does not focus on 2040 values and presents a wide range of historical 2020 values among the models' results for key decarbonization variables such as primary energy and electricity generation. For other major regions, deep decarbonization studies have also been conducted[13–16]. The path to full GHG neutrality in the EU with a focus on 2040 targets, while adhering to realistic starting points and incorporating key current policies already in place such as the tightened $CO_2$ emission standards driving road transport decarbonization, remains underexplored. This near-term realism is crucial to account for lock-ins and trends that significantly influence decarbonization decisions in this and the coming decade.

This study uses the energy-economy model REMIND-EU[9,17–20] to explore cost-efficient pathways towards climate neutrality in the EU under uncertainty regarding key energy system developments. The new pathways provide important insights. First, they allow us to determine overarching emission reduction targets for 2040. Secondly, the in-depth analysis of sectoral transformations by 2040 identifies crucial transformation milestones on the path towards climate neutrality. These sector-and-technology-specific quantitative milestones are valuable reference points for the EU decarbonisation policy framework. Thirdly, a multidimensional sensitivity analysis allows us to identify robust features of the energy transition to climate neutrality as well as key uncertainties. This information is critical to enable policy-makers to develop science-based policies to guide the EU's transformation.

## Results

### EU's emissions on the path to climate neutrality

We explore cost-efficient pathways to achieve climate neutrality in the EU that are consistent with the near-term climate and energy policy framework established by the EU Green Deal. In particular, we consider those policy targets of the Fit-for-55 package and the RePowerEU plans

that are underpinned by concrete measures and firm governance to enforce them. A range of sensitivity scenarios are implemented to provide a robust analysis of the decarbonization trajectories. However, the authors acknowledge that there are numerous political and economic factors that could intervene in this process, providing sources of uncertainty not covered in this analysis. Even achieving 2030 targets will require governments and societies to fully support the current measures and targets set - if there is substantial opposition, or rollback of implemented policies, the 55% emission reduction target will likely not be reached.

Scenario assumptions explore six different dimensions of particular relevance for the EU's energy transition: (1) realised short-term emissions reductions, (2) evolution of final energy demand, (3) availability of sustainable bioenergy, (4) availability of CCS, (5) hydrogen and synthetic fuels availability, and (6) deployment speed of variable renewables. The 336 scenarios developed for this work cover the combinations of these dimensions as listed in Table 1.

The scenarios were calculated using REMIND-EU, an energy–economy–climate multi-regional welfare-optimisation model. It solves for an intertemporal Pareto optimum in economic and energy investments by hard-coupling a Ramsey-type macroeconomic growth model with a technology-detailed energy model, combining the strengths of bottom-up and top-down approaches. It covers all relevant greenhouse gas emitting sectors, as well as options for carbon dioxide removal[21]. It represents in an aggregated way a number of transition-relevant aspects such as technological learning, ageing capital stocks, integration challenges of wind and solar, upscaling challenges of novel technologies, consumer preferences for current technologies, and others.

Our results show that cost-efficient reduction pathways reaching climate neutrality by mid-century are highly convex (Fig. 1a). That is, 2040 total GHG emission (including intra-EU aviation) reductions compared to 1990 would achieve 86% (sensitivity range: 80–93%) (Fig. 1b) by 2040, greatly exceeding the 78% reduction implied by a linear interpolation between the 2030 and 2050 reduction targets.

While this study focuses on the transition itself and not the explicit policies that should govern it, it is clear that stringent climate policies will be needed to achieve this fundamental transformation. The required carbon prices vary substantially depending on the specific scenario settings, but in our reference scenario the required levels to reach climate neutrality are 149 €/tCO$_2$ in 2030, 293 €/tCO$_2$ in 2040, and 411 €/tCO$_2$ in 2050. Further details on mitigation costs are provided in Supplementary Information 1.

Emission reduction potentials vary substantially across sectors, but only one sector stands out as "hard-to-abate": bunker emissions from international aviation and shipping ("Bunkers" in Fig. 1d), which by 2040 are still at 105% of 1990 values, as the shift to biofuels and hydrogen-based fuels barely compensates the growth in demand. The energy supply sector (both electric and non-electric) achieves a 98% reduction in $CO_2$ emissions by 2040, relative to 1990 values. Industry and buildings follow a steady decarbonization trajectory with substantial emission reductions already visible today, while the transport sector currently has higher emissions than 1990 and thus sees a sharp acceleration in emission reductions during the 2030 s. By 2040, residual emissions in demand sectors are reduced to 13% (industry), 27% (buildings), and 22% (transport, excluding international bunkers) relative to 1990 levels. Industrial process emissions remain at 28%, indicating that this sector will require continued mitigation efforts and/or compensation measures beyond 2040.

### Sectoral transformations

Electricity generation is virtually carbon-free by 2040, with coal fully phased out and gas power reduced to 3% (sensitivity range: 0.7% to 15%) of total electricity generation by 2040 (Fig. 2a). This corresponds to a reduction of unabated fossil electricity generation from around

**Table 1 | List of sensitivity dimensions for EU-27 climate neutrality scenarios**

| Type | scenario assumption |
|---|---|
| Short-term policy | 55%, 57% or 59%: level of 2030 GHG emissions reductions relative to 1990. Scope includes Land-use and Land-use change and Forestry (LULUCF) and intra-EU aviation, but excludes extra-EU aviation and international shipping. |
| Efficiency | reference: final energy consumption based on current trends and policies in place.<br>EED 2018 eff: 846 Mtoe (35.4 EJ) final energy by 2030. Final energy consumption compatible with 2018 Energy Efficiency Directive targets.<br>FitFor55 eff: 787 Mtoe (33 EJ) final energy by 2030. Equivalent to 9% reduction by 2030 compared to the 2020 EU reference scenario projections[64] as defined in the Energy Efficiency Directive.<br>RePowerEU eff: 750 Mtoe (31.4 EJ) final energy by 2030. Equivalent to 13% reduction by 2030 compared to the 2020 reference scenario projections as defined in the RePowerEU Plan. |
| Biomass availability | bioLim4: EU-27 biomass availability limited to 4 EJ/yr by 2050 (6 EJ/yr by 2035).<br>bioLim7.5: EU-27 biomass availability limited to 7.5 EJ/yr from 2035 onward.<br>bioLim12: EU-27 biomass availability limited to 12 EJ/yr from 2035 onward.<br>bioLim20: EU-27 biomass availability limited to 20 EJ/yr by 2050: 12 EJ/yr internally produced and, the equivalent of, 8 EJ/yr imported by 2050. |
| Carbon capture and storage | default: annual maximal geological injection rate limited to 0.5% of the total estimated regional storage potential (556 Mt $CO_2$/yr as upper bound)[30]<br>limCCS: annual maximal geological injection rate limited to 0.25% of the total estimated regional storage potential (278 $MtCO_2$/yr as upper bound)<br>unlimCCS: unlimited geological injection rate. |
| Hydrogen and synthetic fuels | default: availability of imports of hydrogen and synthetic fuels to EU-27, reaching 0.6 EJ/yr and 1.75 EJ/yr by 2050 respectively.<br>limH2: lack of internationally traded hydrogen and synthetic fuels availability and reduced hydrogen tax incentives inside the EU. |
| Variable renewables (solar + wind) | default: solar and wind future deployment follow techno-economics parameters and least-cost power generation decision, and upscaling is only limited by adjustment costs on too rapid expansion.<br>limVRE: investment costs for solar and wind technologies are assumed to be 25% higher than in the default scenario.<br>limVRE&Integ.: integration costs for solar and wind are doubled relative to the default scenario, in addition to the 25% investment-cost increase. |

We assume as reference scenario 57% reduction by 2030, reference final energy projections, biomass availability limited to 7.5 EJ/yr, and default CCS, hydrogen and synthetic fuels, and variable renewables assumptions. Details on how these default values were determined are explained in the Methods section under "Scenario design".

40% in 2018–2022, to only 3% (sensitivity range: 1% to 16%) by 2040. Wind and solar technologies provide the majority of the additional electricity generation, following current observed trends (Fig. 2a). Nuclear power is reduced and only contributes 6% (sensitivity range: 6% to 8%) of the total electricity generation, but remains an important contributor in member states that currently rely heavily on nuclear power such as France.

Solar photovoltaics and wind power increase 7-fold (sensitivity range: 4–8-fold) and thus by far dominate electricity generation, providing 2.4 TW (sensitivity range: 1.3 TW to 2.7 TW) of decarbonised electricity supply capacity, and accounting for a combined 79% (sensitivity range: 58% to 82%) of electricity generation by 2040. Our estimates for solar photovoltaic deployment in the reference scenario (0.89TW by 2030 and 1.73TW by 2040, Fig. 2b), go substantially beyond the RePowerEU plan and EU Solar Energy Strategy ambition, but are in line with recent IEA projections[22] and the growth of annual deployment from 2020 to 2023.

Electricity will become the dominant energy carrier until 2040. Its share more than doubles compared to the 2018–2022 average, such that it provides almost half of total final energy demand by 2040 (Fig. 2c). The buildings sector achieves the highest electrification level by 2040, driven by a fast roll out of heat pumps and continued growth of demand from appliances. Its 2040 electrification level is followed closely and soon surpassed by transport electrification, following already-implemented policy frameworks such as CO2 emission performance standards for LDVs and HDVs which result in a rapid increase of electrification.

The challenge of transitioning to a renewables-dominated energy system requires not only replacing existing conventional electricity supply, but also to supply additional electricity for the electrification of hitherto non-electric end uses. Total electricity demand increases by 54% (sensitivity range: 29–61%) compared to the 2018–2022 average until 2040 to reach 332 Mtoe/yr (sensitivity range: 278 Mtoe/yr to 346 Mtoe/yr), and requires a corresponding increase of grid infrastructure.

Furthermore, the increasing share of variable renewable electricity requires the scale-up of electricity storage as well as flexibilization of electricity demand. Especially the widespread deployment of electric vehicles, electricity-based heating as well as hydrogen electrolysers offers new flexibility potentials that can be tapped to balance the grid.

Fossil primary energy demand declines substantially. Coal is fully phased-out in all net-zero scenarios by 2040 (Fig. 3a). The demands for fossil natural gas and oil decline to 39% (sensitivity range: 15% to 89%) and 39% (sensitivity range: 23% to 74%) of today's levels, respectively. These developments imply substantial co-benefits for the European security of energy supply, as can be seen in the resulting trade balance (Fig. 3b). In all net-zero scenarios, the EU by 2040 no longer requires any net imports of coal, while gas and oil imports are reduced by 60% each. Demands for green hydrogen and derived energy carriers such as ammonia and e-fuels grow to 31 Mtoe/yr by 2040 and thus remain much smaller than the reductions in fossil fuel imports. Accordingly, energy security benefits from phasing down fossil imports can be expected to strongly outweigh any new import dependencies even if all e-fuels would be imported.

The transition to climate neutrality results in a contraction of combined fossil and renewable gaseous final energy demand to two thirds of today's levels by 2040, and further down to half by 2050 (Fig. 3c). This is a consequence of the replacement of fossil gas heating systems in buildings and industry by more efficient heat pumps on the path towards climate neutrality. The phase-out of fossil gas imports from Russia and policy measures as part of the RePowerEU package further accelerate these transitions. Increasing demand for hydrogen will only slow the decline, but not reverse the trend of reduced capacity utilisation of the gas infrastructure.

Although far from the 332 Mtoe/yr (sensitivity range: 277 Mtoe/yr to 346 Mtoe/yr) of direct electrification, indirect electrification via hydrogen grows strongly and contributes 29 Mtoe/yr (sensitivity range: 22 Mtoe/yr to 31 Mtoe/yr) of final energy by 2040 mostly used in industry (24 Mtoe/yr, sensitivity range: 19–29 Mtoe/yr). This

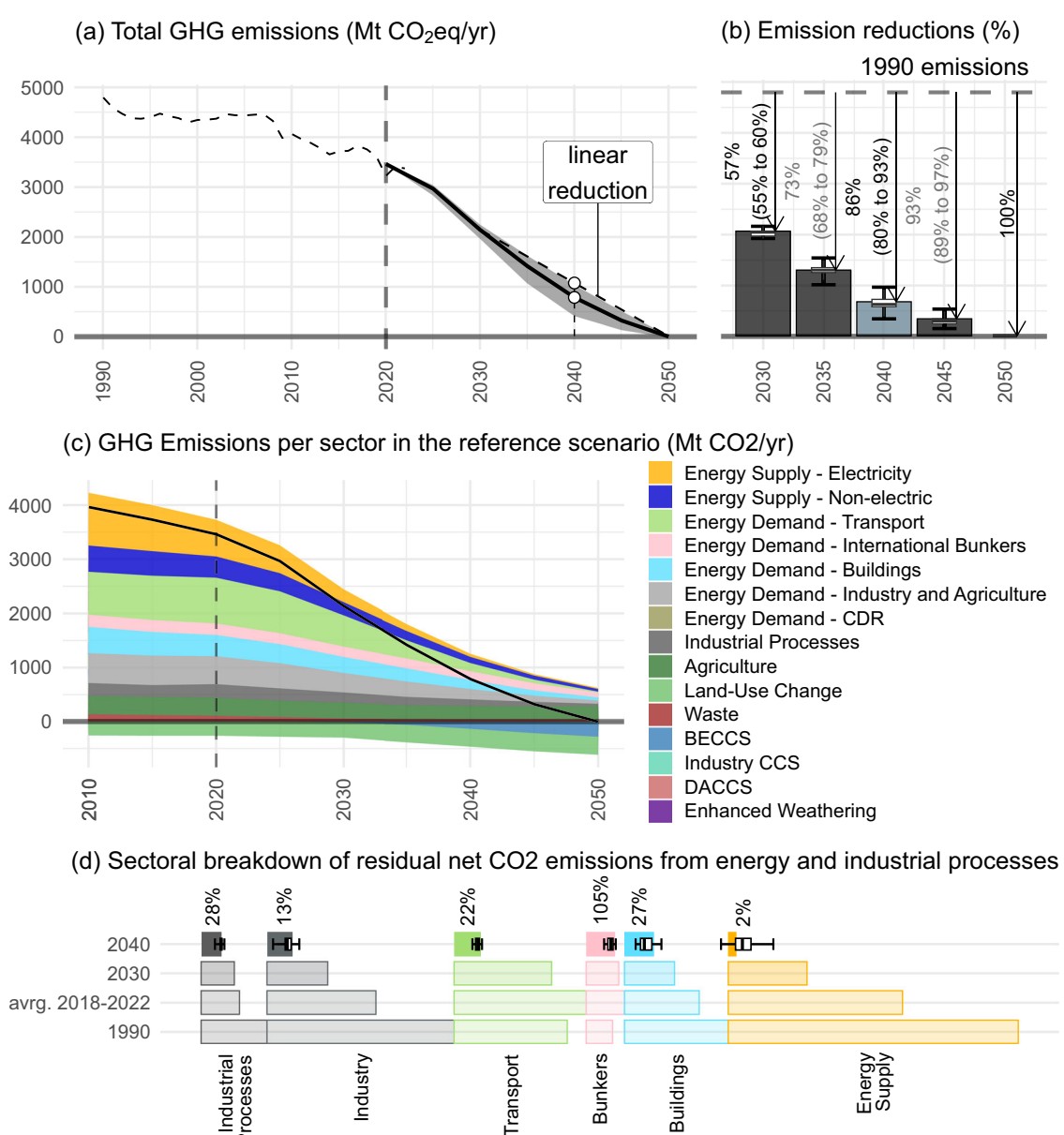

**Fig. 1 | Greenhouse gas emissions for scenarios reaching climate neutrality by 2050. a** Total greenhouse gas (GHG) emissions (Mt $CO_2$eq/yr). Dashed line shows historical emissions; solid line shows the reference scenario (57% reduction by 2030, reference final energy projections, biomass availability limited to 7.5 EJ/yr and default CCS assumption), with shading indicating the range across the full scenario ensemble. **b** GHG emission reductions relative to 1990 (%). Bars indicate reference-scenario values; error bars show the scenario range. Labels denote the minimum–maximum reduction across the ensemble. Reductions are based on GHG emissions including LULUCF and intra-EU aviation for 2030–2040, and including LULUCF and all bunkers for 2045–2050. **c** Sectoral breakdown of gross GHG emissions in the reference scenario (Mt $CO_2$/yr). (**d**) Sectoral breakdown of residual $CO_2$ emissions reduction per decade. Bars show residual emissions by sector relative to 1990 levels; percentages indicate the remaining share of sectoral emissions compared to 1990. Error bars indicate the scenario range. Box plots show the median (centre line), interquartile range (box, Q1–Q3), whiskers extending to minimum and maximum values.

development underscores the importance of an early and ambitious market introduction and scale-up of low-carbon hydrogen. The market ramp-up of green hydrogen in our scenarios, although less ambitious than currently suggested European targets, are in line with recent literature feasibility evaluations[23] (Fig. 3d). E-fuels stay at low quantities (<3 Mtoe/yr) until 2040 and only afterwards become more relevant, growing to 43 Mtoe/yr in 2050, helping to achieve climate neutrality by reducing residual emissions from use of liquids in industry and international transportation.

For the transport sector, the CO2 emissions standards on cars and light commercial vehicles (including the full phase-out of ICE sales by 2035) is the main driver for increasing sales of battery-electric vehicles

and a key step for getting on track for climate neutrality (Fig. 4b). This development implies a more than 2-fold increase of the electricity demand for transportation during the 2030's − 235% increase from 2018–2022 (sensitivity range: 216%–238%) − with a corresponding need for upscaling charging infrastructure.

Beyond LDVs, three main challenges remain in the transport sector: road freight, aviation and shipping decarbonisation (Fig. 4a). Increased range of electric vehicles makes their adoption in road freight segments increasingly competitive, while fuel cell electric vehicles may potentially play a role for longer-haul freight if hydrogen supply is sufficient and cheap. While our scenarios show a continued dominance of fossil fuels for international aviation and shipping

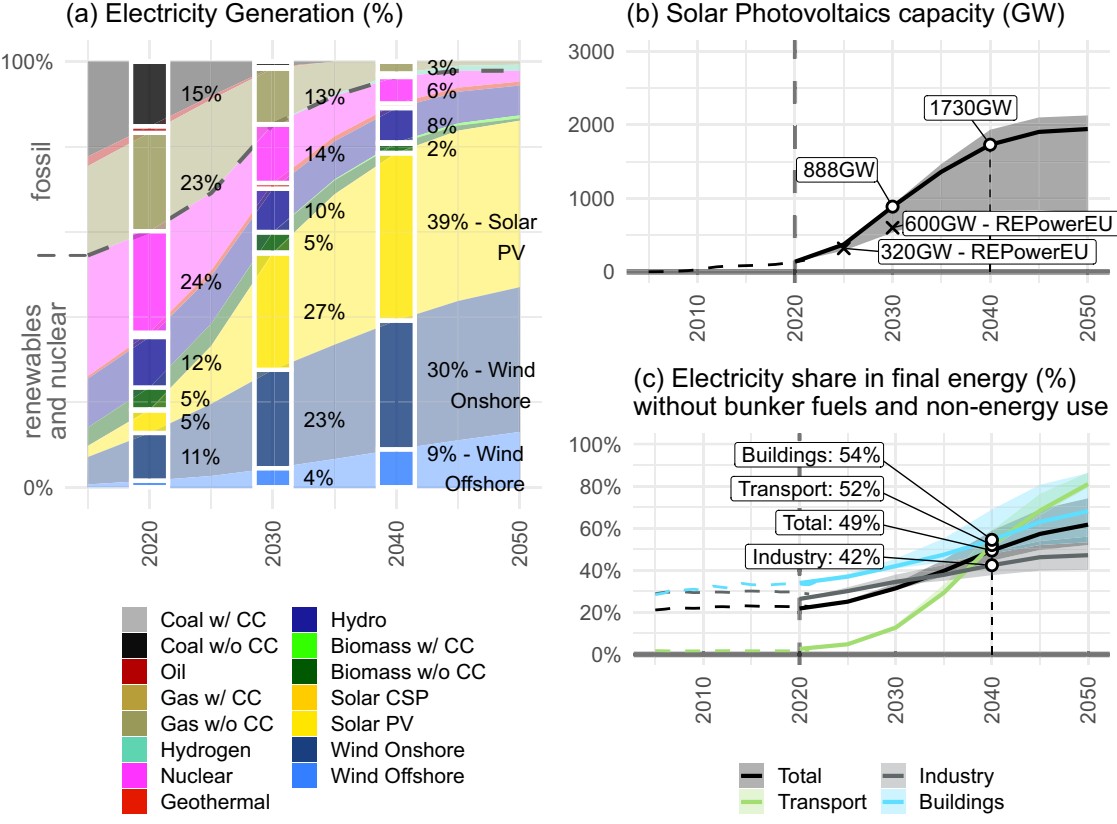

**Fig. 2 | Electricity supply and demand. a** Electricity generation shares by technology in the reference scenario. **b** Upscaling of solar photovoltaics capacity in the reference scenario compared with REPowerEU targets; shaded areas indicate the range across the full scenario ensemble. **c** The transition to electricity as the main energy carrier, shown as the electricity share by sector.

through 2040, these get increasingly replaced by e-liquids and bio-fuel in the following decade (Fig. 4a).

In the buildings sector, a persistent switch towards electricity-based heating, mostly heat pumps, is a robust feature of net-zero pathways. By 2040, this equipment switch has replaced a substantial share of the fossil fuels used for heating today. Decentral electric heating (mainly electric heat pumps) and district heating account for the bulk of the heating demand in 2040 with 66% of useful energy demand in the reference scenario (sensitivity range: 59–84%). The share of useful energy demand for heating provided by liquid or gaseous fuels, currently at more than 50%, is reduced to 29% (sensitivity range: 12–37%) by 2040. Coal heating is completely phased out, while modern biomass-based heating provides the remaining 5% (sensitivity range: 4–5%) of useful energy demand. While the useful energy provided for space and water heating stays relatively constant, the final energy demand to provide this heat decreases by 33% (sensitivity range: 25–58%) between 2020 and 2040, primarily as a result of the switch to heat pumps, which are on average three times as efficient as combustion-based heating.

Given the longevity of industrial installations, major transformation progress is already required by 2040 to put industry on track for climate neutrality. Mitigation strategies vary markedly across industrial sectors. Basic material industries like iron & steel, cement and chemicals account for the bulk of industrial emissions. Recycling of scrap steel via electric arc furnaces supplies one third of the steel demand in 2040, thereby substantially decreasing overall energy demands and increasing electrification rates. Remaining primary steel demands are increasingly produced in hydrogen-based routes, which supply 26% (sensitivity range: 8% to 39%) of the market by 2040. Carbon capture and storage is the dominant mitigation strategy for

cement production as substantial process emissions occur in addition to CO2 emissions from the combustion of fuels. There is only limited scope for direct electrification in the chemical industry, as it relies on hydrocarbons not only for their energy content but also as feedstocks. By 2040, biomass feedstocks, renewable hydrogen, and hydrogen-based e-fuels supply up to one-third of total energy and process chemical sector demands, 20% in the reference scenario (sensitivity range: 17–34%).

Most of the energy demand of industrial subsectors outside bulk material production is either already electric or is required for low-to-medium temperature heat or steam generation and can thus be readily electrified with heat pumps. Thus the share of hydrocarbon fuels in these subsectors drops to around 26% of final energy demand by 2040 in the reference scenario (sensitivity range: 17%–29%).

Due to physical, technical, societal, or political constraints, it is not possible to abate all greenhouse gas emissions. Therefore, carbon sinks are a necessary complement to the phase-out of fossil fuel use. Additionally to the carbon sink of the biosphere (accounted for in the LULUCF sector), further novel carbon dioxide removal (CDR) technologies are necessary to compensate for hard-to-abate emissions (Fig. 4e). Various BECCS technologies provide the majority of total CDR in the scenarios, while DACCS is not used in the reference scenario but comes in as an option in sensitivity scenarios where higher carbon prices are reached. Combining bio-based or synthetic fuels' combustion with CCS furthermore enables CDR in the industry sector.

### Milestones for 2040
Making the European Green Deal policy framework fit for 2040 will require rolling forward targets and expanding the scope of policies to cover more sectors and technologies. Figure 5 presents quantitative

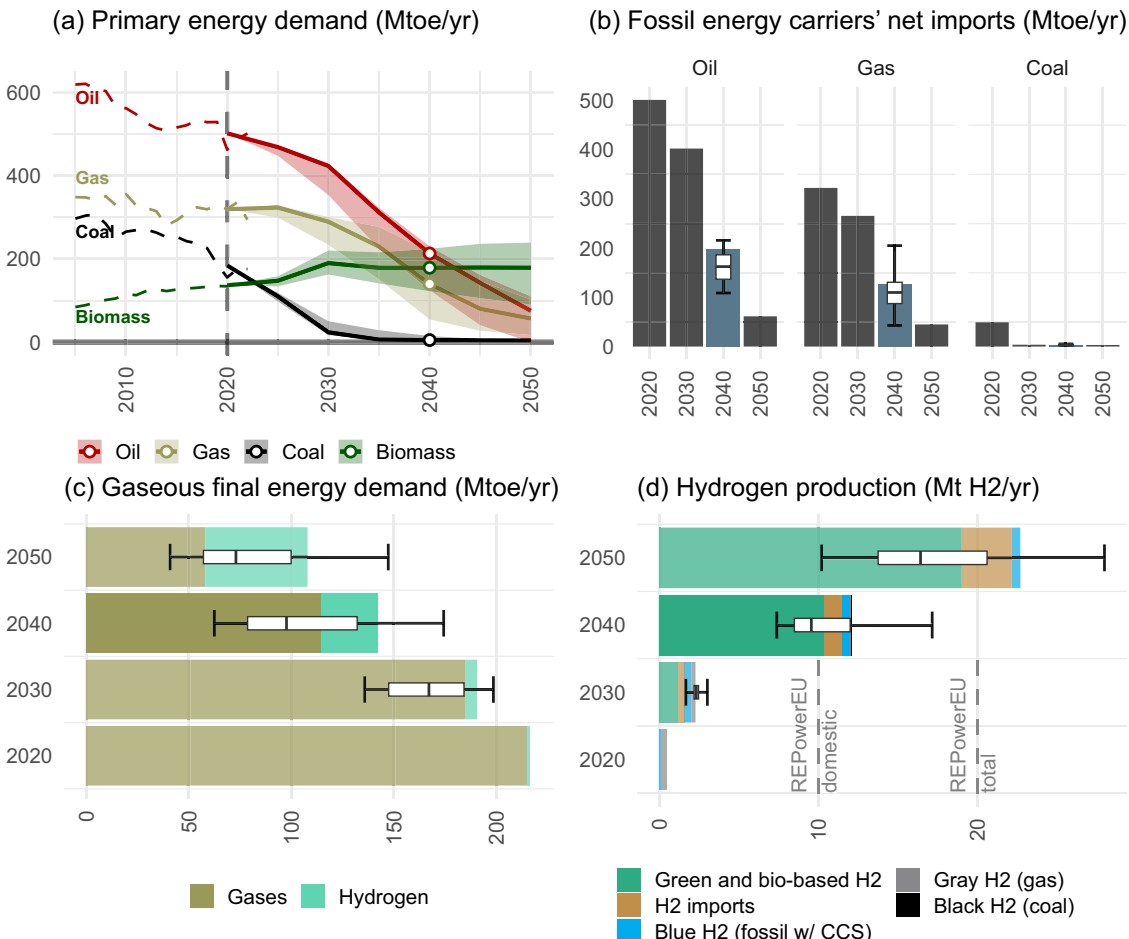

**Fig. 3 | Fuel Supply and Demand. a** Primary Energy demand by fuel. **b** Net imports of fossil energy carriers. **c** Final energy demand for gaseous fuels. **d** Hydrogen production by supply type. Box plots show the median (centre line), interquartile range (box, Q1–Q3), whiskers extending to minimum and maximum values.

estimations derived directly from the scenarios analysed in this paper to inform future targets' decisions. Spreads across scenarios provide an indication of robustness of the results regarding the six uncertainty dimensions described in Table 1.

Our results show that a "linear trajectory linking the Union's climate targets for 2030 and 2040… with the Union's climate-neutrality objective"[24] is not sufficient to achieve cost-efficient reduction pathways to reach climate neutrality by mid-century. A more ambitious 2040 emission reduction target, in between 80% and 93% emission reduction in relation to 1990, is more in line with the replacement dynamics of existing vehicle and heating stocks and thus reduces the amount of stranded assets while promoting early deployment of mitigation options and development of the technologies necessary to achieve climate neutrality by 2050.

Final energy demand is reduced strongly in all our scenarios until 2040, with highest reductions occurring in transport, 46% vs 2018–2022 (sensitivity range: 44–58%), while buildings and industry demand decreases by 22% (sensitivity range: 15–47%) and 20% (sensitivity range: 12–46%), respectively. The main driver for final energy demand reduction is the switch of many combustion-based end-uses to more efficient electricity-based technologies that either minimise energy losses (electric vs combustion-based motors) or, like heat pumps, use ambient heat to increase heating output. Renovation of existing buildings and more efficient vehicles also contribute to reducing final energy demand, but historic experience with accelerating both has been mixed at best[25–27].

Raising buildings' renovation rate is nonetheless important to facilitate the switch to heat-pump based heating systems. The

strengthening of renovation ambition should be targeted at enabling the installation of heat pumps or the connection to green heating grids, i.e. focused on reducing the required flow temperature[28].

The share of renewables in final energy scales up even faster in the next decade, achieving a yearly increase of 5.3%-points (sensitivity range: 4.2–6.7%) in 2030–2040, compared to the 4.5%-points (sensitivity range: 4.1–5%) yearly increase in the current decade. This upscaling will be mainly promoted by the conjunction of carbon-free electricity generation and direct electrification of final energy use. While most of the electricity supply is already decarbonized by 2030, much of the electrification of end-uses is achieved in the 2030-40 period. The share of electricity in final energy increases from 31% (sensitivity range: 31–34%) in 2030 to 49% (sensitivity range: 45–59%) in 2040, a substantial increase compared to the flat 19–20% level throughout the 2010–2020 period. Transport electrification takes the lead on direct electrification due to the fast take-up of battery-electric LDVs.

Recent literature[23] points to the high ambition of the EU's RePowerEU communicated target of 10 million tonnes of domestic renewable hydrogen production by 2030, and a high chance that it will be missed. Our results confirm this finding: despite high uncertainty regarding the long-term hydrogen deployment, all scenarios fall short of meeting the RePowerEU target by 2030. It is only achieved over the course of the 2030 s (Fig. 5), with yearly growth rates of 19% (sensitivity range: 13–20%) in 2030–2040.

Electricity demand will rise substantially until 2040. Solar photovoltaics and wind power will drive the decarbonisation of the sector, increasing four-to-nine-fold and one-and-half-to-three-fold,

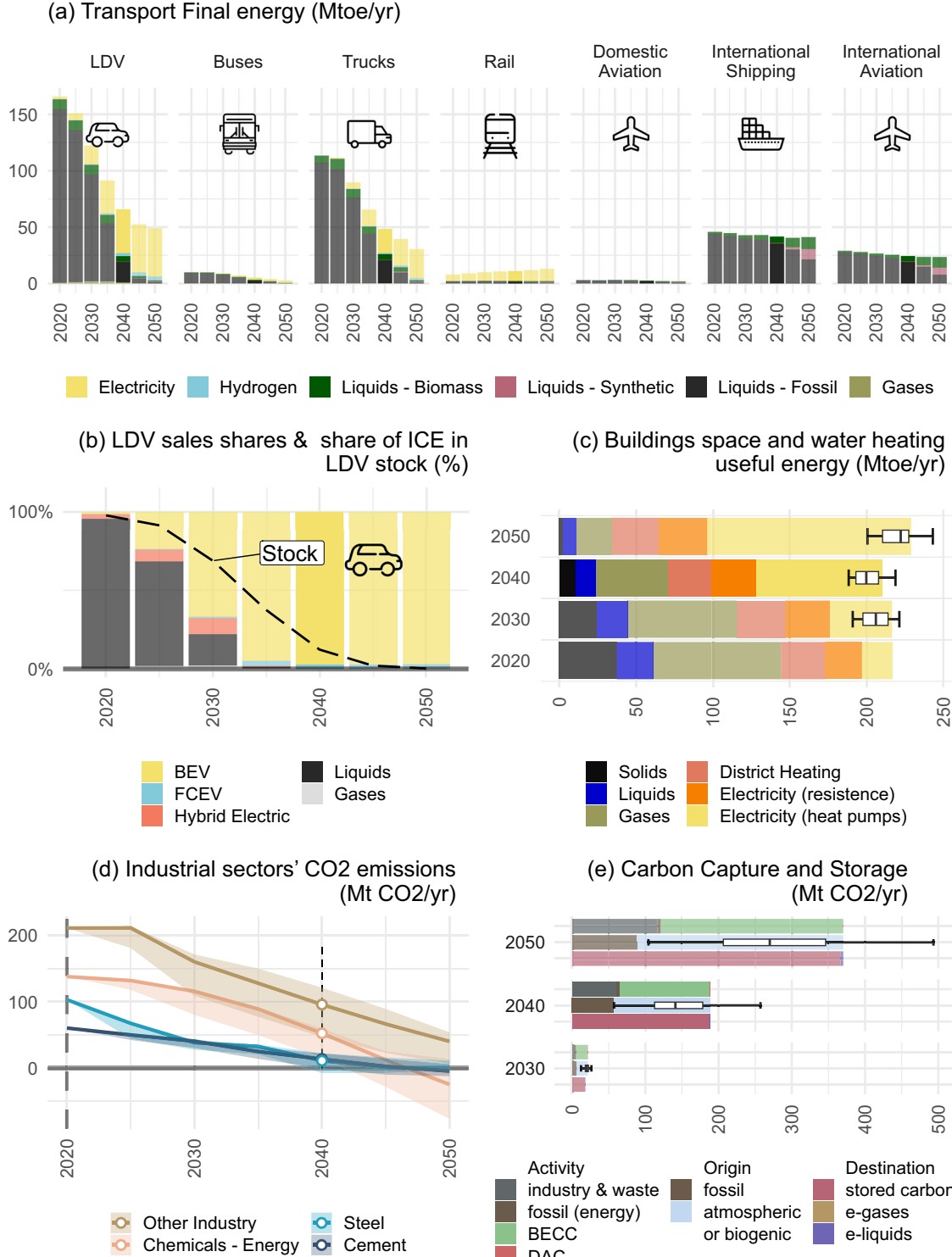

**Fig. 4 | Sectoral transformation indicators. a** Transport final energy; (**b**) Light duty vehicles sales shares (bars) and share of ICEs in total LDV stock (line); (**c**) Useful energy demand for heating in buildings; (**d**) Industry emissions; (**e**) Carbon Capture and Storage by activity (upper bar), origin (middle bar) and by destination (lower bar). Box plots show the median (centre line), interquartile range (box, Q1–Q3), whiskers extending to minimum and maximum values.

respectively, between 2022 and 2040 to provide 79% (sensitivity range: 58–82%) of total electricity generation by 2040. The increasingly dominant share of variable renewable electricity from wind and solar power requires deployment of short- and long-term electricity storage as well as flexibilization of demand. Also, electricity transmission and distribution infrastructure investment and regulation will

need to evolve accordingly to avoid hindering the decarbonisation process.

In all scenarios, some residual fossil emissions, 933 MtCO2 by 2040 (sensitivity range: 531 MtCO2 to 1137 MtCO2) and 376 MtCO2 by 2050 (sensitivity range: 112 MtCO2 to 489 MtCO2), and similarly remaining non-CO2 emissions (e.g. from agriculture) will need to be

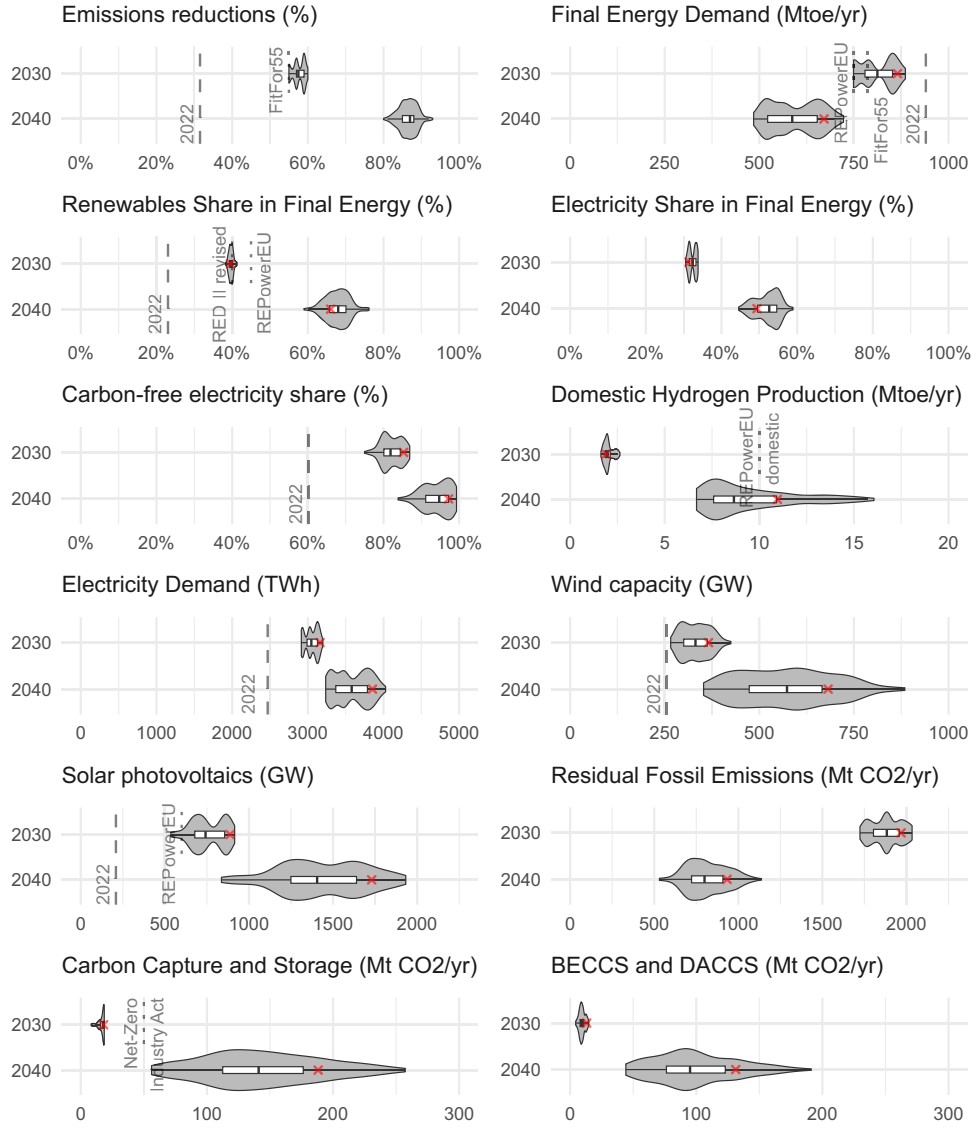

**Fig. 5 | Comparison of key energy transformation indicators including fit-for-55 policy targets for 2030 and 2040.** Red cross symbols represent the results of the reference scenario. Box plots show the median (centre line), interquartile range (box, Q1–Q3), whiskers extending to minimum and maximum values of the sensitivity scenarios, while the shaded areas show the distribution of the scenario results. More information about the chart symbols and indicators can be found at the Supplementary Notes.

offset by negative CO2 emissions. Upscaling of CDR technologies and processes will need to be well underway by 2040 to allow reaching net zero emissions in the following decade. For novel CDR technologies as well as for the abatement of fossil emissions in the industry sector, permanent geological storage of CO2 will be needed. In our net-zero scenarios, the CCS deployment in 2030 ranges between 8–19 MtCO2/yr (18 MtCO2/yr in the reference scenario). While the Net-Zero Industry Act[29] proposes a Union-level objective of a yearly injection rate of 50 MtCO2 to be achieved by 2030, it is unclear if this target will be achieved: Global CCS institute data of all EU27-level storage projects in development or under construction amounts to approximately 14.5 MtCO2/yr by 2030[30], while a 2022 summary of all announced projects by CATF[31] yields a maximum of 43 MtCO2/yr storage capacity in 2030. Considering the long planning periods for site-exploration and project development, these can be considered already ambitious targets to be achieved. The CCS capacity in our scenarios scales to 188 MtCO2/yr in 2040 (sensitivity range: 56 MtCO2/yr to 257 MtCO2/yr). For that, high upscaling rates of 26% (sensitivity range: 16% to 30%)

per year in the period between 2030 and 2040 have to be maintained. As a comparison, offshore wind capacities had annual upscaling rates of 21% in EU27 in the period 2013–2022[32]. These challenges in fast scale-up and large-scale deployment not only apply to the geological storage but also to capture facilities, especially novel CDR technologies like BECCS and DACCS. The scenarios show upscaling rates of 26% (sensitivity range: 21–37%) per year in the period of 2030 and 2040 to reach removal quantities of 131 MtCO2/yr (sensitivity range: 44 MtCO2/yr to 191 MtCO2/yr) in 2040.

## Discussion
This study explores, based on an integrated energy-economy-climate model with high sector detail, pathways to achieve climate neutrality in the EU under uncertainty regarding key energy system developments. Results provide comprehensive insights into the transformation milestones that need to be achieved by 2040 to follow a cost-efficient path towards climate neutrality in 2050, suggesting that emission reductions of 86% (sensitivity range: 80% to 93%) by 2040 relative to

1990, including LULUCF and intra-EU aviation, are consistent with the EU's GHG neutrality goal.

It is important to note that this analysis is grounded in techno-economic optimisation and therefore focuses on least-cost transformation pathways, without incorporating considerations of equity or burden-sharing across countries and world regions. Given the EU's relative affluence and its disproportionate historical contribution to global emissions, accounting for such dimensions would likely imply even higher ambition for a 2040 target[7,33]. In addition, the nominal 2040 reduction is contingent upon the scope in which the target is defined. The EU will only be truly climate neutral if also its share of international aviation and shipping emissions are accounted for. When these sectors are accounted for, the nominal emission reductions achieved in the net-zero scenarios of this study are only 84% (sensitivity range: 80–91%) of 1990, around 2%-points less than under the here-considered emissions scope for the 2030 and 2040 targets. Finally, all results were obtained without relying on emission offsets outside the EU, indicating that external offsets are not required for the Union to reach net-zero.

Economy-wide emission reduction targets alone are insufficient to guide the complex energy transition effort. Sectoral transformation milestones for 2040 play a vital role in aligning the expectations and investments by the various actors needed to achieve mid-century climate neutrality. Our study quantifies important sectoral and technological objectives necessary for the pursuit of the European climate goals. Focus areas include deployment of wind and solar, final energy electrification (e.g. heat pump roll-out), potential fossil carrier bans, renewable technology deployment, decarbonisation of specific industrial sectors, and energy savings targets.

The fundamental systems transformations in our climate neutrality pathways will only be achieved with substantial changes in design and regulation of energy markets. Electricity grids and distribution networks must undergo unprecedented scale-up between 2025 and 2040. Traditional regulation based on quality-of-service should incorporate additional incentives for grid development. Gas network usage will face a strong contraction, and the phase-in of hydrogen as an alternative carrier only manages to slow down the decline, but not stop it. The 2030 s decade will be decisive for bringing novel technologies to wide-scale deployment, such as hydrogen and synthetic fuels, carbon capture and management, or new non-fossil industrial processes. These technologies will be essential in the 2040 s for overcoming the last and most challenging roadblocks on the way to climate neutrality.

While our study provides an understanding of aggregated milestones of the EU-wide transition, further integrated analyses will be needed to guide the transition. Specifically, these should assess EU's decarbonisation in the context of global efforts to achieve the Paris climate change stabilisation targets, in particular with regards to:

- fairness considerations and the interaction of low-carbon transformations in different world regions;
- sector-specific in-depth analysis of opportunities, barriers and achievable pace of the transformation;
- infrastructure requirements and policies for the transition;
- further the understanding of crucial uncertainties and adaptive planning to cope with them;
- the portfolio of policy instruments that are necessary to guide challenging decarbonization activities such as agriculture, specific transportation modes and so on;
- the analysis of geographical differences and acknowledgement of key country-level efforts necessary to achieve the decarbonization goals.

Also, existing studies on the economic and social impacts of this transition and how adverse effects may be addressed[34,35] need to be extended to take the deeper transformation towards 2040 and 2050 into account.

Our results are valuable in making sound long-term plans for decarbonisation until 2040, offering relevant insights for policy-making. Creating a shared vision of the road to climate neutrality among stakeholders on all levels and from all sectors will be paramount for achieving the ground-breaking transformation set out by the EU Green Deal. While many details are still uncertain, the results presented in this study create a consistent basis for critical and constructive discussion to further develop this vision and guide the EU on its path to climate neutrality.

## Methods
### REMIND model

The REMIND-EU[9] model was used to create the 336 scenarios evaluated in this paper. REMIND-EU preserves the global coverage of the REMIND[17,36] model - an open source integrated assessment model widely used in climate assessment literature[18–20] -, and extends its spatial representation by introducing eleven additional European regions to better represent national policies and energy systems.

REMIND is an energy–economy–climate multi-regional intertemporal welfare-optimisation model. It solves for an intertemporal Pareto optimum in economic and energy investments by hard-coupling a Ramsey-type macroeconomic growth model with a technology-detailed energy model, combining the strengths of bottom-up and top-down approaches. It covers all relevant greenhouse gas emitting sectors, as well as options for carbon dioxide removal[21], thus allowing for an integrated assessment of pathways towards climate neutrality. Land-use, agricultural emissions, bioenergy supply and other land-based mitigation options are represented through reduced-form emulators derived from the detailed land-use and agricultural model MAgPIE (Model of Agricultural Production and its Impact on the Environment)[37,38].

REMIND features a high level of detail in the representation of energy-system technologies, trade, and global capital markets. For the scenarios presented here, the model optimises regions individually and uses an iterative adjustment mechanism to clear international markets for (primary) energy carriers and non-energy goods[39].

A short overview of the key components of the model is given in the following paragraphs. The model code is available open source at https://github.com/remindmodel and further documented at https://rse.pik-potsdam.de/doc/remind/3.2.0/.

### Energy system modelling

The energy supply system in REMIND represents the conversion of primary energy carriers into secondary energy carriers and their transport and distribution to end-use sectors. The energy system further accounts for system inertias and path dependencies induced by ageing capital stocks, e.g. in power-plant infrastructure and endogenous learning-by-doing. Additionally, REMIND accounts for challenges related to rapid upscaling of new technologies via cost-markups that are assumed to increase with the square of year-to-year capacity additions[40]. The REMIND model represents the endowments of exhaustible primary energy resources[41] as well as renewable energy potentials based on bottom-up estimates[42,43]. REMIND accounts for cost reductions in solar photovoltaics, concentrating solar power, wind energy and battery storage endogenously via learning-by-doing[44–50]. Technological progress for all other technologies is parameterized via exogenous assumptions.

The REMIND model captures the challenges and options related to the temporal and spatial variability of wind and solar power[42]. In addition to flexible demand response, also inter-regional pooling as well as short-term storage (diurnal time-scales, mostly via batteries) and long-term storage (up to seasonal time-scales) play a key role for facilitating VRE integration. REMIND parameterizes corresponding technology and region-specific VRE storage and grid expansion requirements[51] as well as curtailment rates (i.e., unused surplus share

of VRE electricity generation), which are derived with the help of two detailed electricity production cost models[51,52].

## Energy end uses

An important feature of this study is the representation of demand sectors and cross-sectoral mitigation strategies. In the industry sector, REMIND represents four subsectors: steel, cement, chemicals and other manufacturing. Both primary (virgin) steel production from iron ore and secondary steel production from scrap are represented via a simplified stock-flow-model based on Pauliuk et al.[53]. Energy demand in these subsectors is broken down into heat demands, mechanical work and feedstocks. Mechanical work is already electrified, or can be readily electrified in the future. We further consider indirect electrification of the high-temperature heat inputs for primary steel, cement production and chemical industry via hydrogen. The substitution of heat supply carriers in other manufacturing is represented via a constant elasticity of substitution production function. Feedstocks in the chemical industry must be supplied as hydrocarbon fuels.

Concerning the transport sector, for this study we adopt the coupled system REMIND/EDGE-T[54] to analyse the carriers and transport modes transition towards climate neutrality. Mobility is divided into passenger and freight demands, each broken down by trip length into long-distance and short-medium distance components. The market for each transport demand category is split across different transport modes and vehicle types. Multiple technology options are available for each vehicle type: electricity can be consumed directly in battery-electric cars, buses and trucks, and electric trains. Indirect electrification via hydrogen is available for all road transport options. For passenger cars, mode choice accounts for the value of time of alternative modes. In addition, the technology choice module accounts for dispreferences, e.g., due to range anxiety or low model availability.

Transformation pathways for buildings energy demand are derived from the simulation model EDGE-Buildings. The EDGE-Buildings model projects future floor space for both residential and commercial buildings. For the total building sector, the model then simulates the energy service, useful energy and final energy demands for the end uses (i) space heating, (ii) water heating, (iii) cooking, (iv) space cooling, and (v) appliances and lighting, based on exogenous socio-economic and climate pathways. The REMIND buildings module is then calibrated to the baseline trajectories from EDGE-Buildings, and represents both technology choice and efficiency or behavioural changes as a response to climate policy.

Beyond energy-related $CO_2$, REMIND further represents a wide spectrum of greenhouse gas emissions. $CH_4$ emissions from fossil resource extraction are represented by source. REMIND is coupled to the MAgPIE land use model[37] to derive $CO_2$ emissions from land use, land use change and forestry, as well as $CH_4$ and $N_2O$ emissions from agricultural activities. Abatement options for $CH_4$ and $N_2O$ emissions from energy supply, agriculture, waste and wastewater are based on marginal abatement cost curves from Harmsen et al.[55]. Emissions from fluorinated gases are represented exogenously from Van Vuuren et al.[56]. Emissions from aerosols and short-lived trace gases are based on the GAINS model[57]. Our modelling framework employs the MAGICC[58] reduced complexity climate model to evaluate the resulting changes in global climate variables from the emerging emission scenarios. The impacts of climate change on energy systems and the economy are not considered in the modelling.

## Scenario design

To derive cost-efficient transition pathways, climate policy is represented by assuming an economy-wide CO2-price increasing linearly from today's levels firstly to the levels required to reach the EU green deal 2030 emissions target and later to reach greenhouse gas neutrality in 2050. We further assume that the CO2-price is equal across

sectors. European green deal 2030 policies underpinned by concrete measures and firm governance are embedded in the scenarios definition, and additional measures are assumed to overcome key market failures, such as a phase-out of internal combustion engine vehicle sales in the transport sector as implemented in the Regulation (EU) 2023/851[59].

Six different dimensions are considered for the sensitivity analysis used to identify robust features of the energy transition to climate neutrality as well as key uncertainties: (1) realised short-term emissions reductions, (2) evolution of final energy demand, (3) availability of sustainable bioenergy, and (4) availability of CCS, (5) hydrogen and synthetic fuels availability, and (6) variable renewables penetration.

There is some ambiguity in the interpretation of EU 2030 emissions targets, and corresponding uncertainty about GHG emission reduction levels that will be achieved. According to the weakest interpretation, under the European Climate Law the EU reduces GHG emissions including LULUCF and intra-EU aviation by at least 55% by 2030. However, EU policymakers have implicitly agreed to increase the 2030 nominal reduction target to 57% as part of an agreement to boost natural carbon sinks: With this agreement on natural sinks, there are two overlapping targets - the 55% reduction on GHG emissions, and a target of increasing natural sinks to 310 MtCO2, with the explicit statement that only 225 MtCO2 from these natural sinks may be accounted towards the 55% target. This means that on top of the 55% reduction there will be an additional 85 MtCO2 of negative emissions, bringing the total to 57% reductions[60].

The 59% reduction case is motivated by the possibility that 2030 emissions are further reduced in anticipation of increasingly stringent post-2030 emission constraints, e.g. via banking of permits in the EU-ETS, and/or additional efforts promoted by countries with more stringent 2030 targets, e.g. Germany 65% reduction target.

Energy efficiency improvements greatly reduce the transformation requirements on the supply side. However, its effectiveness is highly dependent on implementation and success of energy efficiency programmes, increased demand side action and cross-sectoral decision making—much of which is under the regulatory authority of the member states. We examine four final energy evolution scenarios to represent the uncertainty realisation of potential efficiency measures in the demand sectors (see Table 1).

The amount of sustainable bioenergy available for the EU is highly uncertain. Following Ruiz et al. ENSPRESO estimations[61], we consider a scenario with bioenergy potential limited to 7.5 EJ/yr, reflecting strict sustainability constraints;[18] a scenario with bioenergy limit of 12 EJ/yr, consistent with more optimistic assumptions about sustainable domestic supply; a high-availability scenario with bioenergy limit of 20 EJ/yr, which includes the equivalent of 8 EJ/yr of imports by 2050, sourced mainly from Latin America and Sub-Saharan African countries[62,63] with an exponential yearly increase rate; and finally, a very-low availability scenario restricting bioenergy potential to 4 EJ/yr to explore a broader sensitivity range.

The capacity for geological carbon storage is highly dependable on the creation of a robust regulatory framework that covers permits, monitoring, cross-border collaboration, local storage acceptance, remuneration and long-term liability, across different European countries of varying legal systems. These regulatory challenges, public acceptance issues as well as technology and cost development give rise to substantial uncertainty. We therefore consider a default reference case with long-term geological injection rates capped at 556 MtCO2/yr; a limCCS case with long-term yearly geological injection rates capped at 278 MtCO2/yr. To explore a broader sensitivity range, we also include a scenario with no explicit limit on geological injection rates.

Hydrogen and synthetic fuels availability could have a key role dealing with hard-to-abate sectors mitigation, however their use demands the maturation of still to be consolidated markets, policies and international trade agreements. We consider two scenarios for

these energy carriers' evolution. The first one examines the availability of hydrogen and synthetic fuels EU-27 imports, equivalent to 14.3 Mtoe/yr and 41.8 EJ/yr by 2050 respectively. Hydrogen imports are mainly sourced from the UK, Norway and Spain, meanwhile e-liquids are mainly sourced from Latin America, Sub-Saharan Africa and Middle-east countries. The second scenario assumes the lack of internationally traded hydrogen and synthetic fuels, together with reduced hydrogen tax incentives inside the EU.

Finally, an additional sensitivity dimension is considered to represent limited variable renewables deployment. The default formulation assumes that future solar and wind deployment follow techno-economics parameters and least-cost power generation decisions based on the default REMIND model and cost assumptions. An alternative case is also considered where investment costs for solar and wind technologies are assumed to be 25% higher than in the default scenario. This additional cost can reduce the economic competitiveness of VRE and slow their deployment relative to the techno-economic baseline. To explore a broader sensitivity range, we include a third scenario that doubles the integration costs for wind and solar while simultaneously applying the limVRE investment-cost increase.

The reference policy scenario assumes a continuation of energy and climate policies currently implemented with 57% emissions reduction by 2030, final energy projections following current observed trends (reference option), biomass availability limited to 7.5 EJ/yr, long-term maximal geological injection rate limited to 556 Mt $CO_2$/yr (default CCS option), availability of hydrogen and synthetic fuels EU-27 imports and variable renewables investments following tecno-economics default assumptions.

## Data availability

A dataset of the modelling results used in this paper is available on GitHub: https://github.com/Renato-Rodrigues/EU_2040_transformation and is archived on Zenodo: https://doi.org/10.5281/zenodo.18661988.

## Code availability

The code to reproduce the experiments used in this paper is available on GitHub: https://github.com/remindmodel/remind/releases/tag/v3.4.0 and is archived on Zenodo: https://zenodo.org/records/14394189. The code to reproduce the results and charts used in this paper is available on GitHub: https://github.com/Renato-Rodrigues/EU_2040_transformation and is archived on Zenodo: https://doi.org/10.5281/zenodo.18661988.

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

## Acknowledgements
This work was supported by the European Union's Horizon 2020 research and innovation programme under grant agreements No 101022622 (ECEMF) and No 101081604 (PRISMA). This study was also supported by the Kopernikus-Ariadne project (FKZ 3SFK5AO-2) funded by the German Federal Ministry of Research, Technology and Space.

## Author contributions
Conceptualisation: R.R., R.P. and G.L. Investigation: R.R. and R.P., Visualisation: R.R. and R.P., Full work analysis: R.R. and R.P., Transport analysis: J.H. and J.M., Buildings analysis: R.H., Industry analysis: M.P., CDR analysis: A.M., Validation: R.R., J.S. and A. M., Policy analysis: R.R., F.S. and J.S., Project administration: R.P., G.L. and L.B., Writing- original draft: R.R., Writing - review: R.P., Writing - editing: all co-authors.

## Funding

## Competing interests
The authors declare no competing interests.
