## [Transparent Peer Review file · Nature Communications]

2040 greenhouse gas reduction targets and energy transitions in line with the EU Green Deal

Corresponding Author: Dr Renato Rodrigues

Version 0:

Reviewer comments:

Reviewer #1

(Remarks to the Author)

This article analyses the net-zero emissions scenarios for the EU using the regional IAM REMIND-EU. It assesses variety of scenarios based on uncertain availabilities on key mitigation options, and identifies sectoral policy instruments to reach net-zero emissions goal of the EU by 2050. Also, this paper is characterized by special focus on energy system transformation in 2040 and discussions on milestones toward net-zero emission target.

Since scientific peer-reviewed literatures on regional net-zero emissions scenarios are still limited today, this paper would provide valuable information and implications for climate policymaking for the EU. Nevertheless, most of energy system implications included in this paper are not surprising because they have already been pointed in other existing literatures. Thus, in the current form, I concern that novelty and contributions to scientific knowledges of this paper are not strong enough. Also, although this paper will be useful for the policymaking in the EU, it would not be interesting for non-EU members or more broader scientific communities very much. In addition, even if publication of this article can be supported with its great contributions on the EU climate policy implications, some information needs to be provided additionally that are necessary for policymakers. Please see following comments for more detail.

First, most of energy system implications included in this paper are already mentioned in the existing literatures. In terms of 2040 emission reduction level, a number of existing literatures have indicated that emission level in cost-efficient emission pathway in 2040 exceed that derived from linear interpolation between near and mid-century emission targets. For example, according to the IPCC-AR6, similar trend is observed in their 1.5C equivalent scenarios. Also, in Rodrigues et al (2022), all models showed similar trends for the EU mitigation scenarios.

<https://doi.org/10.1016/j.energy.2021.121908>

Sectoral heterogeneity on emission reduction level and pace of energy system changes in net-zero emission target is also not surprising. It is already pointed in IPCC-AR5 that electricity decarbonization occurs the fastest. In terms of energy demand sectors, it is well known that each sector has different characteristics, especially so-called difficult to decarbonize sectors, as mentioned in Davis et al.

<https://doi.org/10.1126/science.aas9793>

In terms of sectoral key instruments, Krey et al. (2014) presented that electrification, renewable expansion and CCS are key energy options.

<https://doi.org/10.1007/s10584-013-0947-5>

Also, Iyer et al. (2017) suggested that non-linear energy system transformation is required between near-term and 2050, even when emission pathways are given interpolated. The energy system implications provided in Figure 5 can be useful for policymaking, but its contributions on scientific knowledges is still limited.

<https://doi.org/10.1038/s41558-017-0005-9>

In these regards, this paper needs more comprehensive reviews on existing findings on near to mid term energy system transformation, and to clarify the scientific novelty of this paper in this field.

Second, I acknowledge that the sensitivity analysis on key energy technologies is relatively new aspects in regional scenario assessment compared with existing articles (although there are a number of these exercises by global IAMs...). Nevertheless, there seem several points which need to be reconsidered.

First, whereas this paper assessed multiple scenarios by changing assumptions on key technologies, the uncertainty ranges look too narrow to inform climate policy sufficiently.

For example on bioenergy assumption, according to Rodrigues et al. (2022), biomass penetration of REMIND-EU is the

lowest among three IAMs, while other models shows more optimistic to bioenergy reaching nearly 20 EJ or more by 2050. Given the implications from other IAMs, the uncertainty range used in this study, ranging 7.5-12 EJ/yr, looks unreasonable as a sensitivity analysis to explore robust findings. In addition, according to Daiglou et al. (see below), the EU can be importer of bioenergy in most cases. In this regard, uncertainty ranges need to be reconsidered.

<https://doi.org/10.1007/s10584-020-02877-1>

Also, sensitivity analysis on non-biomass renewables is needed. According to Rodrigues et al. (2022), REMIND-EU largely depends on solar and wind power, whereas other two IAMs seem more conservative on non-biomass renewables. I concur that current situation on solar PV cost supports the trend of future cost declining of solar and wind, but additional assessment based on more comprehensive assumptions are essential for informing policymakers effectively.

In addition, technology uncertainty on mitigation options that are not comprehensively commercialized yet today, such as hydrogen and e-fuels, should be considered, since the current scenario design considers only CCS uncertainty. Also, how can other innovative technologies, such as direct air capture, be considered in sensitivity scenario analysis?

Finally, although the uncertainty ranges are presented with ribbons or error bars in the figures throughout this article, it is important which technology dimensions are impactful to each factor. In this regard, more analysis, such as decomposition analysis with different factors, is needed.

Third, regarding climate policy implications, the information provided by current manuscript looks still insufficient, particularly on detailed sectoral and regional implications. For example, are energy system implications on the buildings sectors common for both residential and commercial sectors? Also, given different EU members have diverse energy and land system characteristics, are the policy implications provided by this paper valid for all member states? Given these points, more detailed sectoral and regional results and their interpretation should be discussed.

In addition, cost implications are totally lacking. As the existing literatures suggested that different technology dimensions could result in completely different mitigation cost and carbon prices implications, this paper does not mention cost and investment aspects at all.

Also, according to the methods, it is mentioned that land use and non-CO₂ emission are covered in this analysis, but no result is provided throughout the article. It is essential to provide simulation results and discussions on non-energy sectors.

My other comments are as follows.

Introduction:

Literature review on research background and motivation are provided, but they are mostly dominated by the European institute. Since there are a number of literatures that focused on near- to mid-century emission pathways for other regions or globally, more comprehensive reviews and emphasizing the originality of this paper are essential. For example, Iyer et al. reported similar analysis for the US.

<https://doi.org/10.1038/s41558-017-0005-9>

Results section

The results shown in the article are highly aggregated, but additional information should be provided to readers and policymakers to understand the scenarios. For example, as sectoral emissions and power generation composition are provided as a share of total numbers in Figure 1 and 2, providing absolute numbers are useful. I presume that such information is included in the dataset provided in the data availability statement, but presenting these data in supplementary figures will be effective.

Fig 1

Only heat and electricity sectors are listed as energy supply. How are the emissions other energy supply sectors reduced? (e.g. coal upgrading, oil refinery, and natural gas processing, hydrogen production). Also, non-energy sector's emission profiles are not found. Please specify.

Line 141

Please specify the carbon source of e-fuel. According to Figure 4e, about a half of captured carbon is derived from fossil sources, and a-third in 2050, which means that e-fuels are no longer carbon neutral if they are converted by fossil-based carbon.

Line 142

Existing literatures suggested that major consumer of e-fuel and hydrogen could be the transport sector, while this study indicated that industry uses most of hydrogen and e-fuels. Why is it not the case for this study?

Figure 4d

More details on industrial sector transformation should be provided. For example for steel production, how much crude steel comes from electric furnaces? How are hydrogen-based reduction and CCS contributed?

Figure 4e

The minimum CCS implementation among the scenarios looks about 200 MtCO₂ in 2050, which is much less than lower constraint of CCS (278 MtCO₂). Why is CCS not fully implemented in this scenario? Please elaborate.

Line 416

Rationale of assuming CCS potential at 556 MtCO₂/yr is unclear. Please specify.

Line 280

Is heat pump installation overlapped with final energy electrification?

Line 371

Are afforestation or reforestation available as CDR sources? Then please provide how much they contribute to net-zero emissions in the mitigation scenarios.

Figure 4c

Does it include both space and water heating?

Figure 5

Please specify what the box and whiskers refer.

Reviewer #3

(Remarks to the Author)

This article provides a highly relevant scenario analysis concerning EU climate policy targets, with specific focus on the forthcoming 2040 interim targets on the path to climate neutrality in 2050. The article makes the central and valid point that intermediate climate targets (i.e. 2030, 2040) are essential to achieve climate neutrality by 2050. It convincingly argues that sectoral and system transformations will be necessary to achieve adequate 2040 targets and climate neutrality by 2040.

From the perspective of a political scientist working on the European Green Deal, several issues emerge in the current version of the article. Already in the abstract, the article assumes that “emission reduction targets and policies up to 2030 are mostly implemented”. While this assumption may be necessary to work on the 2040 targets, there are numerous political and economic factors that are intervening and could intervene by 2030 – new investments in fossil fuels to replace Russian supplies, domestic opposition to the implementation of climate targets in some member states (i.e. Poland), prioritisation of economic recovery over climate targets in a crisis context, failure to roll out renewables as quickly as planned, opposition to large-scale wind power deployment, and so on. The existence of such “intervening variables” could be acknowledged.

Also, while the EU needs to propose 2040 climate targets by 2024, these will most likely be adjusted several times later, as happened with the 2030 targets. A reference to this fluid policy/target process could be included.

In the current form, the article presents the methodology at the end, almost as an appendix to the analysis. While this may be normal for this type of articles, it would be useful to know something about methods earlier on in the article. The REMIND-EU model is mentioned in the introduction, but more information on it could be provided on its most salient features.

Regarding the “four different dimensions of particular relevance for the EU’s energy transition”, how were the four dimensions selected and why? Looking at 2030 emissions reduction targets and energy efficiency targets and projections seems logical, but why is there a focus specifically on biomass (as opposed to other types of renewable energy?) and on CCS?

I also wondered if the scenarios and your model take into account carbon embedded in imports, which is considerable for the EU.

In Figure 5, the units in the axes should be clarified; some graphs report TWh, GW etc. on the y-axis, which instead clearly refers to years. Some of the ranges in the graphs are also unclear and not explained in the text.

Specific in-text comments:

lines 40-41: what is meant by “strengthening” of the Emissions Trading System? This could be explained better;

line 292: what do you mean by “Paris stabilisation targets”; is this a general reference to the climate targets of the Paris agreement, or to something more specific within the treaty?

line 394: the correct name of the law is European Climate Law, not Green Deal climate law.

Reviewer #4

(Remarks to the Author)

This study explores different scenarios for achieving climate neutrality in the EU by 2050 and provides insights into emissions reductions, sectoral transformations, and policy recommendations to support the EU Green Deal's climate goals. It highlights the need for ambitious targets, significant electrification, renewable energy upscaling, and the deployment of Carbon Capture technologies to reach climate neutrality effectively.

The paper is significant for several related fields and stakeholders involved in climate policy, energy transition, and environmental research.

The authors provided an extensive list of references, including various sources from the European Commission, academic papers, research reports, and data sources. These references cover a broad range of topics related to climate change mitigation, energy systems, policy, and environmental modelling.

To my knowledge, the work is original and the methodology is rigorous. Details on the reproducibility of the experiment are provided as online resources.

However, in my opinion, the study fails to mention other important inputs for policymaking that are not only based on technical-economic evidence. While market-driven approaches may effectively regulate industrial emissions, they are often not suited for mitigating pollution from various other sources, including transportation, agriculture, and residential activities. This is why the EU Commission is also planning a Social Climate Fund[1] in the 55% package.

Technical-economic evidence doesn't capture geographical differences: as recently shown in some studies the Member States are progressing toward 2030 objectives and 2050 carbon neutrality at very different paces, showing very different economic decoupling [2]. Settling top-down goals as EU27 is necessary but is not sufficient to achieve carbon neutrality in 2050 or even later[3].

In summary, I believe that the authors' already well-qualified work should be further enhanced by including a brief discussion/mention of socio-economic scenarios, which, in addition to the technical and economic aspects, would offer a more comprehensive view of the significant challenges the EU needs to face toward achieving carbon neutrality.

1. European Commission Social Climate Fund 2021(<https://www.consilium.europa.eu/en/infographics/fit-for-55-social-climate-fund/>).

2. Perissi, I.; Jones, A. Influence of Economic Decoupling in Assessing Carbon Budget Quotas for the European Union. *Carbon Management* 2023, 14, 2217423, doi:10.1080/17583004.2023.2217423.

3. Sgouridis, S.; Carbajales-Dale, M.; Csala, D.; Chiesa, M.; Bardi, U. Comparative Net Energy Analysis of Renewable Electricity and Carbon Capture and Storage. *Nature Energy* 2019, 4, 456–465, doi:10.1038/s41560-019-0365-7

Version 1:

Reviewer comments:

Reviewer #1

(Remarks to the Author)

I appreciate the authors' huge efforts on revision and responses. Although the revised manuscript looks much improved than the initial version, I have some additional comments on the sensitivity analysis.

In comparison with the scenarios presented in Rodrigues et al. (2022), CCS sensitivity ranges seem still too narrow. In terms of the CCS sensitivity scenario, I would suggest adding a scenario with more optimistic assumptions on CCS, because Rodrigues et al. (2022) presented the scenario with CCS (precisely carbon capture) implementation more than 1,000 Gt-CO₂ per year in 2050.

In terms of the Variable Renewables sensitivity case, could you provide a rationale of 25% initial cost penalty, because I could not find any additional information on this assumption?

Also, I presume that 25% cost penalty would not affect the estimated results given the current trend on the drastic cost decline of solar and wind. However, considerable barriers and challenges on VRE upscaling more than 90% of total electricity generation would involve needs for integration measures, such as balancing supply and demand with battery storage or installation of back-up generation resources for a day with low VRE generation due to weather conditions.

In this regard, I expect the authors' to add discussions on this point, or reconsideration of sensitivity ranges as appropriate. Also, more detailed explanations on VREs and their integration measures modeling are expected. As far as I know REMIND is already coupled with detailed power system model (<https://gmd.copernicus.org/articles/16/4977/2023/>), but it is not the case for REMIND-EU?

Regarding the description on Efficiency case in Table 1, would it be possible to provide final energy amount in Exajoule (or Petajoule) for comparison with other figures or tables in this manuscript which are presented in EJ.

(Remarks on code availability)

Reviewer #5

(Remarks to the Author)

Thank you for the revisions to this timely and important manuscript. It comes at a critical juncture in EU climate policy development, as policy and civil society debate on the 2040 greenhouse gas reduction target proposed by the European Commission is in full swing, yet robust quantitative analyses on the proposal - to contrast with the Commission's own impact assessment - are scarce. Hopefully the mainly results have also been fed into the formal consultation process.

I have reviewed the feedback by the previous reviewers, some of which was appropriate, some of which seemed rather extraneous, digressive or tangential. The authors' responses and manuscript modifications are comprehensive, in almost all cases addressing the reviewers' suggestions or criticisms, or in rare cases justifying the decision not to follow the reviewer recommendation (but even then often adding language to clarify a point, or highlight issues not covered by the present manuscript).

I do not see any potential to further improve the manuscript without tradeoffs - e.g. deepening analysis of one aspect and in the process sacrificing focus and depth elsewhere. I also believe further time invested chasing marginal improvements will be outweighed by the missed opportunity to inform a very important policy process with a directly relevant, peer reviewed

quantitative analysis. Hence my recommendation is to accept 'as is'.

(Remarks on code availability)

- Original reviewers' and editor comments are in black font colour.
- Authors' replies to reviewers are in blue font colour.

Contents

Reviewer #1	3
Summary:.....	3
Detailed assessment:	5
Additional comments:	12
Reviewer #3.....	17
Summary:.....	17
Detailed assessment:	17
Additional comments:	21
Reviewer #4.....	22
Summary:.....	22

Reviewer #1

Summary:

This article analyzes the net-zero emissions scenarios for the EU using the regional IAM REMIND-EU. It assesses variety of scenarios based on uncertain availabilities on key mitigation options, and identifies sectoral policy instruments to reach net-zero emissions goal of the EU by 2050. Also, this paper is characterized by special focus on energy system transformation in 2040 and discussions on milestones toward net-zero emission target.

Since scientific peer-reviewed literatures on regional net-zero emissions scenarios are still limited today, this paper would provide valuable information and implications for climate policymaking for the EU.

We thank the reviewer for this positive assessment.

Nevertheless, most of energy system implications included in this paper are not surprising because they have already been pointed in other existing literatures. Thus, in the current form, I concern that novelty and contributions to scientific knowledges of this paper are not strong enough.

Thank you for pointing out that we needed to do a better job explaining the novelty of our work, and the contribution to scientific knowledge.

We accordingly put additional effort in this review into better explaining the relevance of our results and analysis, and how it closes the current knowledge gap.

This can be identified for example, but not exclusively, in the updated introduction paragraph:

“While a few pioneering studies have explored EU deep mitigation pathways⁹⁻¹¹, their results pre-date the establishment of the European Climate Law and its associated policies. One recent model comparison study includes Green Deal targets for 2030 and 2050¹², but the study does not focus on 2040 values and presents a wide range of historical 2020 values among the models’ results for key decarbonization variables such as primary energy and electricity generation. For other major regions, deep decarbonization studies have also been conducted¹³⁻¹⁶. To the best of our knowledge, no peer-reviewed study has analysed the path to full GHG neutrality in the EU with a focus on 2040 targets, while adhering to realistic starting points and incorporating key current policies already in place such as the tightened CO2 emission standards driving road transport decarbonization. This near-term realism is crucial to account for lock-ins and trends that significantly influence decarbonization decisions in this and the coming decade.”

Moreover, this study provides a crucial counterpoint to the European Union commission's impact assessment on a 2040 climate target, extending the range of evaluated sensitivity analysis and assessments to sustain more informed decarbonization policy decisions. In this regard, we thank the reviewer comments underlining the need for additional sensitivity analysis dimensions, that we hope to have covered in this revised version of the paper.

We adjusted the manuscript to better emphasise this, like in the introduction paragraphs:

“The European Climate Law mandates the establishment of an intermediary climate target for 2040, whose legislative proposal is scheduled for 2025^{3,6}. This target-setting process has been substantially informed by the European Scientific Advisory Board on Climate Change⁷, which has drawn heavily from preliminary results of the scenarios developed in this paper. As was the case with the EU 2030 targets, science-based analysis is an important enabler of informed decision-making in the process of setting and revising climate targets and policies. The current study aims to contribute to this process by providing an independent counterpoint to the European Commission's impact assessment on the 2040 climate target⁸, extending the range of evaluated sensitivities to support robust policy decisions.

Setting an aggregated EU-wide emission reduction target is an important, but not sufficient, step to achieve climate neutrality by 2050. The proposed 2040 emission target will likely be accompanied by sectoral transformation milestones, underpinned by policies and measures similar to those defined for 2030 as part of the Fit-for-55 package. Defining these targets requires detailed quantitative modelling of the transformation across all sectors, considering also relevant uncertainties.”

Also, although this paper will be useful for the policymaking in the EU, it would not be interesting for non-EU members or more broader scientific communities very much.

We agree that the highest relevance of our paper will be for European policymakers and scientists. However, we are convinced that it is also quite relevant for a global audience. The European results presented in this paper can serve as both a counterpoint and a guiding framework for orienting the decarbonization process in other countries and regions of the World outside the EU. The EU's leading role in decarbonization discussions, its significant global economic influence, and its investment and development strategy being rooted in its decarbonization strategy, exemplified by the European Green Deal relevance, will likely influence the strategies of other regions worldwide.

The EU's success or failure in achieving its mitigation targets will have profound global effects, shaping the ambition and targets of other regions in the effort to tackle climate change. To address the important point raised by the reviewer, and to clarify the relevance of the EU climate mitigation analysis to global climate change mitigation goals, we have included an additional paragraph at the beginning of the paper, underscoring the EU's significance in this context:

“The European Union (EU) is distinguished by its ambitious climate mitigation targets, positioning itself as a leader in global climate action. As a region that constitutes nearly one-

sixth of the global economy, the EU's decarbonization strategy is poised to exert significant global influence. The EU's commitments and policies are expected to have far-reaching effects, inspiring and influencing the climate ambitions and actions of other regions worldwide.”

In addition, even if publication of this article can be supported with its great contributions on the EU climate policy implications, some information needs to be provided additionally that are necessary for policymakers. Please see following comments for more detail.

Detailed assessment:

First, most of energy system implications included in this paper are already mentioned in the existing literatures. In terms of 2040 emission reduction level, a number of existing literatures have indicated that emission level in cost-efficient emission pathway in 2040 exceed that derived from linear interpolation between near and mid-century emission targets. For example, according to the IPCC-AR6, similar trend is observed in their 1.5C equivalent scenarios. Also, in Rodrigues et al (2022), all models showed similar trends for the EU mitigation scenarios.

<https://doi.org/10.1016/j.energy.2021.121908>

Although previous analyses have observed some of the characteristics present in our results, they do not invalidate the need for an updated formulation and revision of the scenarios to yield more robust conclusions.

For instance, the mentioned study by Rodrigues et al. (2022) pre-dates the establishment of the Green Deal and its associated policies. This reason alone could be considered as sufficient to assume its results are not suitable for drawing robust conclusions about intermediate climate targets, in which short-term influence and path dependency are crucial. As a direct consequence, they would be insufficient to provide reliable milestones for achieving the 2040 decarbonization targets as intended by our work. Moreover, Rodrigues et al. allowed a 5% deviation as an acceptable neutrality target and only covered CO₂ emissions, which are acceptable assumptions in longer term evaluations, but are too lenient for abiding by EU legislation and addressing the short-term decarbonization targets examined in this paper. Our study adopts much stricter conditions to achieve emission reduction targets and covers the full spectrum of GHG emissions to accurately assess the EU decarbonization goals.

Similarly, the IPCC AR6 results are also affected by these issues. Additionally, most 1.5°C equivalent scenarios lack the rigorous treatment necessary to reflect historical tendencies, starting points, policies in place, and geographical compatibility with the EU27. To illustrate this fact, the European Scientific Advisory Board on Climate Change evaluated over 1100 emission scenarios, including those from IPCC AR6, and only 63 scenarios passed their vetting process for the 2040 target analysis. Of these, 36 scenarios were selected for the final report, with the majority being represented by preliminary results of the scenarios developed in this paper.

To better clarify our contributions in the paper, we have summarised these points in the introduction section, highlighting the significance of this paper's findings.

Reference: European Scientific Advisory Board on Climate Change. Scientific Advice for the Determination of an EU-Wide 2040 Climate Target and a Greenhouse Gas Budget for 2030–2050. <https://data.europa.eu/doi/10.2800/609405> (2023).

Sectoral heterogeneity on emission reduction level and pace of energy system changes in net-zero emission target is also not surprising. It is already pointed in IPCC-AR5 that electricity decarbonization occurs the fastest. In terms of energy demand sectors, it is well known that each sector has different characteristics, especially so-called difficult to decarbonize sectors, as mentioned in Davis et al.

<https://doi.org/10.1126/science.aas9793>

In terms of sectoral key instruments, Krey et al. (2014) presented that electrification, renewable expansion and CCS are key energy options.

<https://doi.org/10.1007/s10584-013-0947-5>

Also, Iyer et al. (2017) suggested that non-linear energy system transformation is required between near-term and 2050, even when emission pathways are given interpolated.

<https://doi.org/10.1038/s41558-017-0005-9>

We agree with the reviewer's observation that our results do not contradict previous research in these areas. We would like to emphasise that our contribution lies not in invalidating previously investigated decarbonization analysis, but rather in providing concrete values for where each sector should be to achieve the European decarbonization goals and intermediate targets. Thus providing the necessary scientific background on which policy targets can be defined.

As mentioned in the previous comment, additional content was added to the paper introduction to better clarify the research gap that this paper intends to fulfil and highlight the relevance of this paper's findings.

The EU strongly emphasises the importance of science-backed targets, which can only be established through comprehensive scientific analysis of various pathways. Our work aims to meet this need by delivering quantification of detailed sectoral targets that underpin the broader decarbonization strategy.

As mentioned previously, results that pre-date the establishment of the European Climate Law and recent significant geopolitical developments, which have substantially influenced European energy and decarbonization policies, have limited applicability to shorter-term target analysis. Furthermore, the advancements in scientific methods over the past ten years applied to the methods used in these simulations warrant considerable recognition, serving to consolidate and extend the conclusions from previously published studies.

The energy system implications provided in Figure 5 can be useful for policymaking, but its contributions on scientific knowledges is still limited.

In these regards, this paper needs more comprehensive reviews on existing findings on near to mid term energy system transformation, and to clarify the scientific novelty of this paper in this field.

The key transformation indicators presented in Figure 5 were selected to represent critical figures that assist not only in policymaking but primarily in quantifying the scale, variation, and uncertainty of these dimensions. This is intended to support the future development of various areas, such as:

- Technological feasibility studies grounded in real-world scale requirements.
- Sector-specific studies with cross-sectoral effects estimated in this research.
- Policy instrument design impacting multiple energy carriers, sectors, and the overall mitigation strategy.

More specifically, integrated assessment results of the required variable renewables shares and the electrification share of the economy, compatible with a successful European mitigation strategy, can serve to dimension grid investment studies; publications assessing supply chain requirements for batteries or other flexibility providers; future research on the establishment of direct and indirect electrification trade patterns,...

Final energy demand estimations could be used to, for example, compare a carbon-free high GDP region against future pressures from developing countries' catching-up policies.

The design of the electricity sector suitable for different levels of carbon-free electrification shares can be put up to test in more technical and theoretical publications. The data on remaining residual fossil fuels can be used to dimension carbon dioxide removal studies, or face specific industrial sector studies against the expected decarbonization effort required from them.

The quantification of technology-specific deployment rates, such as those for wind, solar, CCS, and BECCS, are crucial for publications focused on learning-by-doing, investment bottlenecks, and technical feasibility assessments.

All the above are examples of possible contributions that this study could provide to scientific oriented knowledge.

Second, I acknowledge that the sensitivity analysis on key energy technologies is relatively new aspects in regional scenario assessment compared with existing articles (although there are a number of these exercises by global IAMs...). Nevertheless, there seem several points which need to be reconsidered.

First, whereas this paper assessed multiple scenarios by changing assumptions on key technologies, the uncertainty ranges look too narrow to inform climate policy sufficiently.

We thank the reviewer for acknowledging the contribution made by the sensitivity analysis carried out in our work. More importantly, we greatly appreciate the reviewer's detailed analysis on gaps that could improve this sensitivity exercise.

We went into extensive work to cover the recommendations made by the reviewer in this regard, further extending this aspect of the paper and improving the robustness of the results presented.

The original assessed 36 scenarios now comprise a total of 192 scenarios due to this exercise. You can see more details about the additions in the following comments.

For example on bioenergy assumption, according to Rodrigues et al. (2022), biomass penetration of REMIND-EU is the lowest among three IAMs, while other models shows more optimistic to bioenergy reaching nearly 20 EJ or more by 2050. Given the implications from other IAMs, the uncertainty range used in this study, ranging 7.5-12 EJ/yr, looks unreasonable as a sensitivity analysis to explore robust findings. In addition, according to Daioglou et al. (see below), the EU can be importer of bioenergy in most cases. In this regard, uncertainty ranges need to be reconsidered.

<https://doi.org/10.1007/s10584-020-02877-1>

Following the reviewer's suggestion, additional sensitivity scenarios were added to reflect more extreme bioenergy availability cases.

A more optimistic bioenergy availability scenario was added, where EU-27 has available up to 20 EJ/yr biomass primary energy by 2050. This scenario considers the equivalent of 8 EJ/yr bio-liquids directly imported by the EU by 2050, which can be crucial to decarbonize hard-to-abate industrial activities and transportation modes. This bio-energy flow is mainly sourced from Latin America and Sub-Saharan African countries due to their biomass potentials, following Wu et al. (2019) and Daioglou et al. (2020), with an exponential yearly increase rate until 2050.

Moreover, following recently observed extreme events that directly affect biomass availability, an additional scenario portraying a future with even further limited biomass availability was also added to the sensitivities evaluated in this work. This scenario considers a linear reduction of available biomass in EU-27 dropping from 6 EJ/yr biomass primary energy by 2035 to 4EJ/yr biomass availability by 2050.

All scenarios developed in this work take advantage of the global aspect of REMIND IAM formulation and also consider internationally traded primary energy biomass flows, endogenously reflecting the important point raised by the reviewer. The EU acts as an importing biomass region in all scenarios evaluated.

<https://onlinelibrary.wiley.com/doi/10.1111/gcbb.12614>

<https://doi.org/10.1007/s10584-020-02877-1>

Also, sensitivity analysis on non-biomass renewables is needed. According to Rodrigues et al. (2022), REMIND-EU largely depends on solar and wind power, whereas other two IAMs seem more conservative on non-biomass renewables. I concur that current situation on solar PV cost supports the trend of future cost declining of solar and wind, but additional assessment based on more comprehensive assumptions are essential for informing policymakers effectively.

We agree with the reviewer's statement that current cost and deployment trends of PV align well with the tendencies for declining future costs of solar and wind. We would like to also mention that the observed accelerating growth of wind and especially solar deployment over the last three years ("The exponential growth of solar power will change the world" - The economist, 2024; "Clean energy is entering the energy system at an unprecedented rate, including more than 560 gigawatts (GW) of new renewables capacity added in 2023" - World Energy Outlook 2024, IEA) are well in line with previous REMIND scenarios the near-term developments observed in the current scenarios.

In contrast, early CCS deployments assumed by other IAMs and previous studies with more conservative non-biomass renewables penetration were not realised, or can be considered unreasonable to be reached in the short-term, which limits potential competition for VRE.

Following the reviewer's suggestion, an additional sensitivity scenario was added to the paper to create a counterfactual representing limited variable renewables deployment. Under this new sensitivity scenario, investment costs for solar and wind technologies are assumed to be 25% higher than in the default scenario.

In addition, technology uncertainty on mitigation options that are not comprehensively commercialized yet today, such as hydrogen and e-fuels, should be considered, since the current scenario design considers only CCS uncertainty.

We agree that the availability of emerging technologies such as hydrogen and e-fuels is a critical uncertainty in designing an effective decarbonization strategy, especially given their crucial role in mitigating emissions in hard-to-abate sectors.

Following the reviewer's suggestion, an additional sensitivity scenario dimension was added to reflect these energy carriers availability and deployment in time. There are two hydrogen and e-fuels availability cases represented in our current simulations. The first one, that is now considered as our new default formulation, examines the availability of hydrogen and synthetic fuels EU-27 imports, equivalent to 0.6 EJ/yr and 1.75 EJ/yr by 2050 respectively. Hydrogen imports are mainly sourced from the UK, Norway and Spain, meanwhile e-liquids are mainly sourced from Latin America, Sub-Saharan Africa and Middle-east countries. The second scenario assumes the lack of internationally traded hydrogen and synthetic fuels, together with reduced hydrogen tax incentives inside the EU.

We went into careful consideration with sectoral experts before deciding to update our reference scenario definition, considering the additional sensitivity scenarios included in this revision. The reference scenario was adjusted to better reflect current tendencies for these energy carriers. This update to the reference scenario largely explains the differences between the charts in our initial submission and those in the current revision. The previous reference scenario remains as part of our scenario ensemble.

Also, how can other innovative technologies, such as direct air capture, be considered in sensitivity scenario analysis?

Direct air capture is an available mitigation technology present in all our scenarios, and it is only competitive enough as a viable 2040 mitigation alternative in the more stringent biomass availability scenario (bioLim 4 in Supplementary figure 7).

Additional figures and comments were added to the supplementary material underlying these results and the relationship between bioenergy availability, BECCS and DAC requirements (see Supplementary Information 1.E Carbon Dioxide Removal).

Finally, although the uncertainty ranges are presented with ribbons or error bars in the figures throughout this article, it is important which technology dimensions are impactful to each factor. In this regard, more analysis, such as decomposition analysis with different factors, is needed.

We agree that further decomposition analysis would enrich the paper contents, however due to length limitations it is quite challenging to take this up in the paper.

As already mentioned, an additional supplementary information section was added to the manuscript to present additional and more detailed results of the simulations carried out in this paper.

Future research can be developed to further analyse and extend the key transformations underlined in this paper.

Third, regarding climate policy implications, the information provided by current manuscript looks still insufficient, particularly on detailed sectoral and regional implications. For example, are energy system implications on the buildings sectors common for both residential and commercial sectors?

We recognize the relevance of providing deeper analysis of each of the sectors' decarbonization strategies, however due to length and scope limitations it is impossible to provide extensive regional and sectoral details, which in our opinion deserve dedicated publications by themselves.

For a full-system cross-country study like this, we think it is appropriate to consider the residential and commercial subsectors together as one building sector. In both subsectors, energy demand is dominated by heating and electric appliances and lighting, so their decarbonisation is driven by

the combination of electrifying the heating demand and decarbonising electricity supply. We acknowledge that both subsectors have different characteristics (e.g. in the potential for heating grid connection or ownership structures) that can be very relevant in more detailed analyses of the building sector which could build on our full-system scenarios.

We have taken your comment as an opportunity to change the paragraph on the building sector modelling in the methodology section to be more precise on the sub-sectoral detail.

Also, given different EU members have diverse energy and land system characteristics, are the policy implications provided by this paper valid for all member states?

Given these points, more detailed sectoral and regional results and their interpretation should be discussed.

Once again we recognize the relevance of the point raised by the reviewer but we have to deal with size and scope limitation decisions.

The analysis of the paper focuses on EU-27 results, and although results for specific countries and regions within EU-27 are also provided in the supplementary data provided with the paper, there is both a format and message limitation to fulfil the reviewer request.

We opted to provide an analysis more focused on high-level insights focused on EU-27 as a whole, but we recognize the relevance of the point raised by the reviewer. In order to achieve a suitable compromise, an additional section was added to the supplementary information of the paper (see Supplementary Information 1.B. Regional results), including additional figures and brief comments with regional specific results, as well as underlining the need to follow up analysis diving further in country specific dynamics and results.

In addition, cost implications are totally lacking. As the existing literatures suggested that different technology dimensions could result in completely different mitigation cost and carbon prices implications, this paper does not mention cost and investment aspects at all.

Additional figures and comments were added to the supplementary information including mitigation costs and implicit carbon price information (see Supplementary Information section 1.A Mitigation costs).

The supplementary data provided with the paper also include additional variables reflecting scenario specific technology and mitigation cost components, and the open-source REMIND code include bottom-up technology cost parameters information.

Also, according to the methods, it is mentioned that land use and non-CO₂ emission are covered in this analysis, but no result is provided throughout the article. It is essential to provide simulation results and discussions on non-energy sectors.

We appreciate the reviewer's valuable observation. In response, we have updated Figure 1c to provide a detailed representation of emissions in the paper's reference scenario, explicitly covering the full scope of total GHG emissions—including non-CO₂ gases (e.g., CH₄, N₂O) from both energy and non-energy sectors.

A dedicated section in the Supplementary Information (Section 1.D) was also included presenting results for GHG emissions per gas type, and additional sector-specific charts and comments were added in the Supplementary Information (Section 1.C) to provide energy and non-energy use per sector and fossil content.

Given the manuscript's focus on overarching targets and system-wide transformation pathways, scope and length constraints limited our ability to provide in-depth analysis of all individual sectors in the main text. We fully acknowledge the relevance of these sectors, and we agree that more detailed analysis would merit dedicated studies.

Additional comments:

My other comments are as follows.

Introduction:

Literature review on research background and motivation are provided, but they are mostly dominated by the European institute. Since there are a number of literatures that focused on near-to mid-century emission pathways for other regions or globally, more comprehensive reviews and emphasizing the originality of this paper are essential. For example, Iyer et al. reported similar analysis for the US.

<https://doi.org/10.1038/s41558-017-0005-9>

This paper focuses on European Union (EU) results, rather than conducting a comparative analysis of mid-term decarbonization strategies across various global regions. For this reason the literature review is centred around EU-specific recent literature.

In response to reviewer comments, we have included in the introduction additional references to specific decarbonization analysis centred on Nationally Determined Contributions (Iyer *et al.*, 2017), United States (Bistline *et al.*, 2023), China (Liu *et al.*, 2021) and industrialised economies (Schreyer *et al.*, 2020). This aims to better situate our analysis within a broader global context.

We understand the relevance of writing papers comparing mid-term strategies between different world regions (and have done so), but this would go much beyond the scope of this paper and would force us to reduce detail in other parts and thus become less relevant to the EU audience.

References:

Iyer, G. et al. Measuring progress from nationally determined contributions to mid-century strategies. *Nature Climate Change* 7, 871–874 (2017).

Bistline, J. et al. Emissions and energy impacts of the Inflation Reduction Act. *Science* 380, 1324–1327 (2023).

Schreyer, F. et al. Common but differentiated leadership: strategies and challenges for carbon neutrality by 2050 across industrialized economies. *Environmental Research Letters* 15, 114016 (2020).

Liu, Z. et al. Challenges and opportunities for carbon neutrality in China. *Nat Rev Earth Environ* 3, 141–155 (2021).<https://www.nature.com/articles/s41558-017-0005-9>

Results section

The results shown in the article are highly aggregated, but additional information should be provided to readers and policymakers to understand the scenarios. For example, as sectoral emissions and power generation composition are provided as a share of total numbers in Figure 1 and 2, providing absolute numbers are useful. I presume that such information is included in the dataset provided in the data availability statement, but presenting these data in supplementary figures will be effective.

We have added sectoral emissions (Figure 1c) as well as power generation and capacity composition (Supplementary Figures 3 and 4) to the manuscript.

The information about data availability can be found in the manuscript section, Data and code availability:

“A dataset of the modelling results and the code to reproduce the results and charts used in this paper is available on GitHub: https://github.com/Renato-Rodrigues/EU_2040_transformation under a CC-BY-4.0 licence.”

Fig 1

Only heat and electricity sectors are listed as energy supply. How are the emissions other energy supply sectors reduced? (e.g. coal upgrading, oil refinery, and natural gas processing, hydrogen production). Also, non-energy sector’s emission profiles are not found. Please specify.

We thank the reviewer for this important observation. Figure 1d has been updated to reflect aggregated energy supply emissions, now including additional energy supply sectors such as oil refining, natural gas processing, coal transformation, and hydrogen production. We chose to present energy supply emissions in aggregated form to enhance readability and reduce visual complexity in the main figure. More detailed sectoral breakdowns are available throughout the manuscript, in the supplementary information, and in the provided dataset including all paper results.

Regarding non-energy sector emissions, those from industrial processes are included under “industrial processes” in Figure 1d. Emissions from waste and agriculture for the reference scenario are provided in Figure 1c.

Line 141

Please specify the carbon source of e-fuel. According to Figure 4e, about a half of captured carbon is derived from fossil sources, and a-third in 2050, which means that e-fuels are no longer carbon neutral if they are converted by fossil-based carbon.

The reviewer's question raises the issue of e-fuel accounting. For total emissions, the following two scenarios are identical:

- 1) Captured atmospheric/biogenic CO₂ is primarily used for e-fuels such that they are carbon neutral with the consequence that less will be available to be stored geologically (higher abatement, lower CDR).
- 2) E-fuels are produced with a share of fossil captured carbon but therefore more of the atmospheric carbon gets stored geologically (lower abatement, higher CDR).

The issue you raise thus only has an impact on the reported sectoral split of emissions, eg “how much do emissions from bunkers go down”, “how much CDR takes places”.. The REMIND model does not separately track for which purpose CO₂ captured from a specific source is used, thus the share of fossil captured CO₂ that goes into e-fuels and geologic storage is identical. This follows the logic of scenario 2.

Figure 4.e was adjusted to include information about the proportion of captured CO₂ that is directed to e-fuels production. As can be seen, only a very small fraction of the non-fossil based captured carbon is used for e-fuels production. Although other scenarios present different shares of e-fuel production, their CO₂ demand never exceeds the supply of atmospheric/biogenic CO₂.

Line 142

Existing literatures suggested that major consumer of e-fuel and hydrogen could be the transport sector, while this study indicated that industry uses most of hydrogen and e-fuels. Why is it not the case for this study?

In 2040, the available e-fuel quantities are still very low, and the available hydrogen is in higher demand from the industry as most gas-based industry processes can easily shift to using hydrogen, whereas the use of hydrogen for shipping and aviation is expected to face larger hurdles.

By 2050, with larger e-fuel quantities, also these transport modes make up a relevant share of e-fuel+hydrogen demands.

Figure 4d

More details on industrial sector transformation should be provided. For example for steel production, how much crude steel comes from electric furnaces? How are hydrogen-based reduction and CCS contributed?

As the figures are already quite complex, we added the information in the industry-related paragraph that electric arc furnaces supply one-third of steel demand by 2040 in the industry-related sector, while hydrogen-based routes supply roughly 26%.

“Recycling of scrap steel via electric arc furnaces supplies one third of the steel demand in 2040, thereby substantially decreasing overall energy demands and increasing electrification rates. Remaining primary steel demands are increasingly produced in hydrogen-based routes, which supply 26% of the market by 2040.”

Figure 4e

The minimum CCS implementation among the scenarios looks about 200 MtCO₂ in 2050, which is much less than lower constraint of CCS (278 MtCO₂). Why is CCS not fully implemented in this scenario? Please elaborate.

The limited CCS scenario constraint is an upper bound, and not a lower bound, of the total possible amount of carbon injection per year in the scenarios.

This bound is not necessarily binding in all scenarios, as CCS is not enforced but it is instead dependent on the cost-optimal decision to build CCS capacity among other alternative mitigation alternatives.

To illustrate this, the additional sensitivities added in this revision push even further the minimal amount of carbon capture and storage needed to achieve EU climate policy ambitions. If the EU reaches 57% emission reductions by 2030 and implements such strong efficiency policies in line with Re-PowerEU energy efficiency savings ambitions it could achieve its 2050 net-neutrality goal with as low as 112 MtCO₂ of CCS capacity according to our scenarios.

Line 416

Rationale of assuming CCS potential at 556 MtCO₂/yr is unclear. Please specify.

CCS annual injection rate limits in our scenarios are defined as proportions of the total regional storage potential, which is based on Global CCS Institute data (<https://co2re.co/StorageData>). The rationale is to reflect physical limitations on safe maximum injection rates into the reservoir.

The default scenario assumes an upper bound for annual injection rates corresponding to 0.5% of the total storage potential available, meanwhile the limited CCS scenario assumes half of this capacity.

Additional text was added to the paper, in Table 1, to include this information.

Line 280

Is heat pump installation overlapped with final energy electrification?

Thanks for this comment. We agree that this overlaps and have changed the sentence to avoid this.

Line 371

Are afforestation or reforestation available as CDR sources? Then please provide how much they contribute to net-zero emissions in the mitigation scenarios.

Afforestation and reforestation are included as part of the Land-Use and Land-Use Change (LULUCF) emissions. LULUCF as well as agricultural emissions are emulated in REMIND by marginal abatement cost curves derived from the land-use model MAgPIE.

Figure 4c

Does it include both space and water heating?

Figure 4c has been updated to include the useful energy provided by all relevant energy carriers for buildings' space and water heating. The title of the figure has been updated accordingly to reflect this information.

Figure 5

Please specify what the box and whiskers refer.

Additional information has been added to the figure label and the Supplementary Notes section to clarify the figure elements used in the paper.

Reviewer #3

Summary:

This article provides a highly relevant scenario analysis concerning EU climate policy targets, with specific focus on the forthcoming 2040 interim targets on the path to climate neutrality in 2050. The article makes the central and valid point that intermediate climate targets (i.e. 2030, 2040) are essential to achieve climate neutrality by 2050. It convincingly argues that sectoral and system transformations will be necessary to achieve adequate 2040 targets and climate neutrality by 2040.

We thank the reviewer for this positive assessment.

Detailed assessment:

From the perspective of a political scientist working on the European Green Deal, several issues emerge in the current version of the article. Already in the abstract, the article assumes that “emission reduction targets and policies up to 2030 are mostly implemented”. While this assumption may be necessary to work on the 2040 targets, there are numerous political and economic factors that are intervening and could intervene by 2030 – new investments in fossil fuels to replace Russian supplies, domestic opposition to the implementation of climate targets in some member states (i.e. Poland), prioritisation of economic recovery over climate targets in a crisis context, failure to roll out renewables as quickly as planned, opposition to large-scale wind power deployment, and so on. The existence of such “intervening variables” could be acknowledged.

We thank the reviewer for the very relevant comment. We fully agree that achieving the 2030 targets is not a given - elections may bring governments to power that want to stop climate mitigation such that existing policies may be weakened or cancelled, further geo-political challenges may arise that will put even more pressure on budgets, etc. We are in this study only analysing the techno-economic transition and how it might play out given continued support

Additional text was added to the paper acknowledging the possibility of “intervening variables” that could hinder the decarbonization transition.

“We explore cost-efficient pathways to achieve climate neutrality in the EU that are consistent with the near-term climate and energy policy framework established by the EU Green Deal. In particular, we consider those policy targets of the Fit-for-55 package and the RePowerEU plans that are underpinned by concrete measures and firm governance to enforce them. **A range of sensitivity scenarios are implemented to provide a robust analysis of the decarbonization trajectories. However, the authors acknowledge**

that there are numerous political and economic factors that could intervene in this process, providing sources of uncertainty not covered in this analysis. Even achieving 2030 targets will require governments and societies to fully support the current measures and targets set - if there is substantial opposition, or roll-back of implemented policies, the 55% emission reduction target will likely not be reached.”

Also, while the EU needs to propose 2040 climate targets by 2024, these will most likely be adjusted several times later, as happened with the 2030 targets. A reference to this fluid policy/target process could be included.

Additional text was added to the introduction section referencing this paper potential contribution to possible upcoming climate targets revision processes.

“The European Climate Law mandates the establishment of an intermediary climate target for 2040, whose legislative proposal is scheduled for 2025^{3,6}. This target-setting process has been substantially informed by the European Scientific Advisory Board on Climate Change⁷, which has drawn heavily from preliminary results of the scenarios developed in this paper. **As was the case with the EU 2030 targets, science-based analysis is an important enabler of informed decision-making in the process of setting and revising climate targets and policies.** The current study aims to contribute to this process by providing an independent counterpoint to the European Commission's impact assessment on the 2040 climate target⁸, extending the range of evaluated sensitivities to support robust policy decisions.”

In the current form, the article presents the methodology at the end, almost as an appendix to the analysis. While this may be normal for this type of articles, it would be useful to know something about methods earlier on in the article. The REMIND-EU model is mentioned in the introduction, but more information on it could be provided on its most salient features.

Nature communications publication format recommends describing the methodology in its own section by the end of the manuscript as observed by the reviewer.

Following the reviewer suggestion, an additional paragraph was added to the main paper text describing briefly the model and its main features:

“The scenarios were calculated using REMIND-EU, an energy–economy–climate multi-regional welfare-optimisation model. It solves for an intertemporal Pareto optimum in economic and energy investments by hard-coupling a Ramsey-type macroeconomic growth model with a technology-detailed energy model, combining the strengths of bottom-up and top-down approaches. It covers all relevant greenhouse gas emitting sectors, as well as options for carbon dioxide removal²⁵. It represents in an aggregated way a number of transition-relevant aspects such as technological learning, ageing capital

stocks, integration challenges of wind and solar, upscaling challenges of novel technologies, consumer preferences for current technologies, and others.”

Regarding the “four different dimensions of particular relevance for the EU’s energy transition”, how were the four dimensions selected and why? Looking at 2030 emissions reduction targets and energy efficiency targets and projections seems logical, but why is there a focus specifically on biomass (as opposed to other types of renewable energy?) and on CCS?

In response to the reviewer’s comment, we have incorporated two additional sensitivity dimensions in this revision, focusing on variable renewables and the development of hydrogen and e-fuels. This enhancement aims to improve the robustness of the paper’s results and address uncertainty surrounding key decarbonization energy carriers.

Additional details on these sensitivity dimensions have been included in the *Methods* section, specifically within the *Scenario Design* subsection. Below, we provide a brief summary of the key rationale behind the selected sensitivity dimensions:

1. 2030 Target:

Clearly, the 2030 targets are the linchpin of the short-to-medium climate mitigation efforts in the EU. Although the actual policies were designed with this target in mind, they do not necessarily ensure hitting the target exactly, but can lead to some over/under-achievement. It is important to test how 2040 emission reductions on the way to climate neutrality will be influenced by such over/underachievement in 2030.

2. Energy Efficiency:

Efficiency improvements strongly interact with other mitigation policies and can potentially advance mitigation efforts by several years. According to a recent estimation by the ESABCC and our own research, the currently-implemented efficiency targets will be very challenging to achieve. Given that the governance of the EED is not very strong, the level of efficiency improvements actually reached in 2030 will depend on the strength of policies enacted in the Member States. It is thus advisable to explore a range of efficiency improvements in our sensitivity analysis.

3. Biomass Availability:

Biomass plays a dual role in the energy transition by substituting fossil fuels to lower emissions and by enabling negative emissions when combined with carbon capture and storage (BECCS). However, biomass availability is constrained due to land-use limitations, and its sustainability depends on strict land-use regulations. Without adequate regulation, biomass expansion could cause adverse ecological impacts, including displacement of agricultural production, leading to direct and indirect land-use change emissions [Heck et al., 2018] and displacement effects of agricultural production might cause significant direct and indirect land-use change emissions, substantially reducing the climate benefit of biomass [Merfort et al., 2023]

Heck, V., Gerten, D., Lucht, W. et al. Biomass-based negative emissions difficult to reconcile with planetary boundaries. *Nature Clim Change* 8, 151–155 (2018). <https://doi.org/10.1038/s41558-017-0064-y>

Merfort, L., Bauer, N., Humpenöder, F. et al. Bioenergy-induced land-use-change emissions with sectorally fragmented policies. *Nat. Clim. Chang.* 13, 685–692 (2023). <https://doi.org/10.1038/s41558-023-01697-2>

4. CCS Feasibility:

The rapid upscaling of CCS to climate-relevant levels remains uncertain. Past experiences, including the first wave of CCS projects from 2010 to 2017, highlight significant implementation gaps, with only a fraction of the planned global capacity (~10 Mtpa CO₂ out of ~120 Mtpa CO₂) being realized (<https://www.catf.us/resource/policy-framework-for-carbon-capture-and-storage-in-europe>). Additionally, CCS faces social acceptance challenges due to concerns over perceived risks and potential misuse by the fossil fuel industry to delay decarbonization [d'Amore et al., 2020]. To reflect these uncertainties, we include a low CCS-availability sensitivity scenario.

d'Amore, F., Lovisotto, L., Bezzo, F.. Introducing social acceptance into the design of CCS supply chains: A case study at a European level, *Journal of Cleaner Production*, Volume 249, 2020, 119337, ISSN 0959-6526, <https://doi.org/10.1016/j.jclepro.2019.119337>.

5. Hydrogen and E-Fuels:

Hydrogen and synthetic fuels (e-fuels) are key components of deep decarbonization strategies, particularly in hard-to-abate sectors. Their future development, however, depends on overcoming challenges related to production costs, infrastructure, and market uptake. Including this sensitivity dimension allows us to better capture uncertainties in the scaling of these technologies.

6. Variable Renewables (VRE):

The integration of variable renewable energy (VRE) sources, such as wind and solar, is crucial for the energy transition. Evaluating VRE availability sensitivities provides valuable insights into worst-case scenarios, helping to assess the robustness and adaptability of decarbonization pathways.

I also wondered if the scenarios and your model take into account carbon embedded in imports, which is considerable for the EU.

Additional information has been added to the Supplementary Notes section (SI2) to clarify this point.

“Our results account for nationally originated emissions and international transport bunker emissions. Future research could include embedded carbon in imports and policies, such as carbon border adjustments, to assess the decarbonization potential of traded goods.”

In Figure 5, the units in the axes should be clarified; some graphs report TWh, GW etc. on the y-axis, which instead clearly refers to years. Some of the ranges in the graphs are also unclear and not explained in the text.

Information about the x-axis units of the charts presented in figure 5 were moved to the chart titles to improve readability.

Additional information has been added to the figure label and the Supplementary Notes section to clarify the figure elements used in the paper.

Additional comments:

Specific in-text comments:

lines 40-41: what is meant by “strengthening” of the Emissions Trading System? This could be explained better;

Additional text was added to the manuscript exemplifying a measure of strengthening of the EU Emissions Trading System as part of the Fit for 55 policy package:

“These include, among others, **the extension of the EU Emissions Trading System to new sectors and the tightening of its caps**, the increase of renewable energy sources in the overall energy mix, the increase of energy efficiency targets, as well as the introduction of emissions reduction targets for cars and vans.”

line 292: what do you mean by “Paris stabilisation targets”; is this a general reference to the climate targets of the Paris agreement, or to something more specific within the treaty?

Additional text was added to the manuscript to clarify that we refer to limiting climate change effects according to be aligned with the Paris agreement ambitions.

line 394: the correct name of the law is European Climate Law, not Green Deal climate law.

We thank the reviewer for the correction and we adjusted the paper content accordingly.

Reviewer #4

Summary:

This study explores different scenarios for achieving climate neutrality in the EU by 2050 and provides insights into emissions reductions, sectoral transformations, and policy recommendations to support the EU Green Deal's climate goals. It highlights the need for ambitious targets, significant electrification, renewable energy upscaling, and the deployment of Carbon Capture technologies to reach climate neutrality effectively.

The paper is significant for several related fields and stakeholders involved in climate policy, energy transition, and environmental research.

The authors provided an extensive list of references, including various sources from the European Commission, academic papers, research reports, and data sources. These references cover a broad range of topics related to climate change mitigation, energy systems, policy, and environmental modelling.

To my knowledge, the work is original and the methodology is rigorous. Details on the reproducibility of the experiment are provided as online resources.

We thank the reviewer for this positive assessment.

However, in my opinion, the study fails to mention other important inputs for policymaking that are not only based on technical-economic evidence. While market-driven approaches may effectively regulate industrial emissions, they are often not suited for mitigating pollution from various other sources, including transportation, agriculture, and residential activities. This is why the EU Commission is also planning a Social Climate Fund[1] in the 55% package.

We agree with the point raised by the reviewer comment, however due to scope and length limitations this research focused on providing a techno-economic analysis that can serve as basis for future research more focused on policy instruments portfolios more suited to guide this decarbonization process.

We have included an additional sentence to the conclusions section, in the future research paragraph, to underline the point raised by the reviewer:

“While our study provides an understanding of aggregated milestones of the EU-wide transition, further integrated analyses will be needed to guide the transition. Specifically, these should assess EU’s decarbonisation in the context of global efforts to achieve the Paris climate change stabilisation targets, in particular with regards to:

- ● fairness considerations and the interaction of low-carbon transformations in different world regions;

- sector-specific in-depth analysis of opportunities, barriers and achievable pace of the transformation;
- infrastructure requirements and policies for the transition;
- further the understanding of crucial uncertainties and adaptive planning to cope with them;
- **the portfolio of policy instruments that are necessary to guide challenging decarbonization activities such as agriculture, specific transportation modes and so on;**
- the analysis of geographical differences and acknowledgement of key country-level efforts necessary to achieve the decarbonization goals.”

Technical-economic evidence doesn't capture geographical differences: as recently shown in some studies the Member States are progressing toward 2030 objectives and 2050 carbon neutrality at very different paces, showing very different economic decoupling [2]. Settling top-down goals as EU27 is necessary but is not sufficient to achieve carbon neutrality in 2050 or even later[3].

We agree that considering geographical differences are a prerequisite to design successful decarbonization policies at European Union level.

This is a general principle reflected directly in some already existent directives. For example, the Emission Trading System (ETS) targets are fully tradable within the EU, meanwhile Effort Sharing Regulation (ESR) targets include country level differentiation. These mechanisms provide the possibility of countries making up for underachievement of other countries.

We also agree with the reviewer's point that top-down goals as EU-27 is a necessary but not sufficient condition to achieve the decarbonization goals. Additional content was added to the paper introduction to reflect this.

But, although not sufficient, setting an EU-wide target is key to this process. This is even more clear when we consider that National Energy and Climate Plans (NECPs) and Long-Term Strategies should be evaluated and commented on by the EU commission to make them align with the EU-wide targets.

This work focuses on high-level insights of the EU-wide target, but this should not come in detriment of further analysis on countries' specificities and a clear acknowledgement that country-level efforts are key to achieve decarbonization goals. In order to make direct reference to this point in the paper, an additional future research line was included in the conclusions section of the paper:

“...the analysis of geographical differences and acknowledgement of key country-level efforts necessary to achieve the decarbonization goals.”

Moreover, the simulations created for this study included country and regional differentiation within the EU that range beyond techno-economics, including behavioural assumptions, specific legislations and societal trends. Due to scope and length limitations, a more detailed analysis of geographical differences cannot be addressed by this research alone. Additional content was added to the supplementary information material, mainly in the regional results section, to underline some considered geographical differences and some potential fields for future research in this subject.

In summary, I believe that the authors' already well-qualified work should be further enhanced by including a brief discussion/mention of socio-economic scenarios, which, in addition to the technical and economic aspects, would offer a more comprehensive view of the significant challenges the EU needs to face toward achieving carbon neutrality.

We thank the reviewer again for the kind words for our work. We agree that the question of socio-economic impacts and interactions are of high relevance, but beyond the scope of our paper. We thus added: "Also, existing studies on the economic and social impacts of this transition and how adverse effects may be addressed^{36,37} need to be extended to take the deeper transformation towards 2040 and 2050 into account." to the discussion section in the hope to have given enough emphasis in our manuscript for the importance and the need to account for these dimensions in the successful design and implementation of the EU decarbonization policy.

[36] Vandyck, T., Della Valle, N., Temursho, U., Weitzel, M., 2023. EU climate action through an energy poverty lens. *Sci Rep* 13, 6040. <https://doi.org/10.1038/s41598-023-32705-2>

[37] Weitzel, M., Vandyck, T., Rey Los Santos, L., Tamba, M., Temursho, U., Wojtowicz, K., 2023. A comprehensive socio-economic assessment of EU climate policy pathways. *Ecological Economics* 204, 107660. <https://doi.org/10.1016/j.ecolecon.2022.107660>

- Original reviewers' and editor comments are in black font colour.
- Authors' replies to reviewers are in blue font colour.

Contents

Reviewer #1	3
Remarks to the Author:.....	3
Reviewer #5.....	7
Remarks to the Author:.....	7

Reviewer #1

Remarks to the Author:

I appreciate the authors' huge efforts on revision and responses. Although the revised manuscript looks much improved than the initial version, I have some additional comments on the sensitivity analysis.

We thank the reviewer for their contribution during this process that helped us to improve the paper!

In comparison with the scenarios presented in Rodrigues et al. (2022), CCS sensitivity ranges seem still too narrow. In terms of the CCS sensitivity scenario, I would suggest adding a scenario with more optimistic assumptions on CCS, because Rodrigues et al. (2022) presented the scenario with CCS (precisely carbon capture) implementation more than 1,000 Gt-CO₂ per year in 2050.

Following the reviewer's suggestion, we added an additional sensitivity scenario with unlimited geological CO₂ injection rates to our scenario ensemble to represent a more favourable CCS setting. Interestingly, this assumption has only a limited impact on capacity deployment outcomes. By contrast, carbon price levels in the more stringent scenarios are substantially affected, primarily reflecting the mitigation alternative costs required to meet the 2050 target.

Several factors explain why our findings differ from those in Rodrigues et al. (2022):

- Stronger competition from alternative mitigation options: Solar and wind costs have continued to fall rapidly, reinforcing their role as the backbone of the energy transition and reducing the relative attractiveness of CCS (Ember, 2025).
- Improved understanding of electrification potential: Recent literature (Zhang et al., 2025, Link et al, 2024, Luderer et al., 2022) and real-world developments highlight faster progress in sectoral electrification, e.g., accelerated EV manufacturing, battery cost and autonomy improvements, and expanding portfolios of zero-carbon industrial technologies.
- Persistent implementation challenges for CCS (Global CCS Institute, 2023): Repeated cost overruns, cancellations, and delays in carbon-capture projects reduce the modelled competitiveness of large-scale CCS deployment.
- Tighter constraints on sustainable biomass supply (): biomass availability and the growing value of alternative biomass uses limit the feasible scale of BECCS.

Taken together, these developments substantially narrow the techno-economic space for large-scale CCS on the road to climate neutrality in the EU compared with earlier assessments. Even under optimistic injection-rate assumptions, CCS does not scale to the levels seen in Rodrigues et al. (2022), mainly because competing mitigation options have become significantly more cost-effective and scalable.

Ember. (2025). The Electrotech Revolution. Ember Energy.

Luderer, G., Madeddu, S., Merfort, L. et al. Impact of declining renewable energy costs on electrification in low-emission scenarios. *Nat Energy* 7, 32–42 (2022). <https://doi.org/10.1038/s41560-021-00937-z>

Zhang, S., Zhao, P., Wang, F. et al. Feasibility, cost and decarbonization potential of clean pathways for heavy-duty road transportation. *Nat. Rev. Clean Technol.* 1, 846–860 (2025). <https://doi.org/10.1038/s44359-025-00119-0>

Link, S., Stephan, A., Speth, D. et al. Rapidly declining costs of truck batteries and fuel cells enable large-scale road freight electrification. *Nat Energy* 9, 1032–1039 (2024). <https://doi.org/10.1038/s41560-024-01531-9>

Global CCS Institute. Facilities Database. <https://co2re.co/StorageData> (2023).

In terms of the Variable Renewables sensitivity case, could you provide a rationale of 25% initial cost penalty, because I could not find any additional information on this assumption?

Thank you for this question. The 25% investment-cost increase for solar and wind reflects a conservative yet plausible range of cost escalation that could slow renewable deployment. It captures potential short-term spikes driven by supply-chain disruptions, raw-material price volatility, higher financing costs, and policy or trade uncertainties. This penalty was selected based on modeller judgement of recent policy changes and cost fluctuations in recent years.

Also, I presume that 25% cost penalty would not affect the estimated results given the current trend on the drastic cost decline of solar and wind. However, considerable barriers and challenges on VRE upscaling more than 90% of total electricity generation would involve needs for integration measures, such as balancing supply and demand with battery storage or installation of back-up generation resources for a day with low VRE generation due to weather conditions.

In this regard, I expect the authors' to add discussions on this point, or reconsideration of sensitivity ranges as appropriate. Also, more detailed explanations on VREs and their integration measures modeling are expected. As far as I know REMIND is already coupled with detailed power system model (<https://qmd.copernicus.org/articles/16/4977/2023/>), but it is not the case for REMIND-EU?

Thank you for your comment. We fully agree with you that at high VRE shares, integration costs are more relevant than direct technology costs. An additional sensitivity scenario was introduced to represent higher variable renewable energy (VRE) integration costs. In the limVRE&Integ scenario, solar and wind integration challenges are doubled relative to the default case, in addition to the 25% investment-cost increase already assumed in the limVRE scenario.

These increased integration challenges have quite some impact on the mix of VRE: solar power reduces by 31%, less variable wind offshore increases by 4%, and battery storage capacities increase by 8% to ~700GW/4900 GWh in the reference scenario. Overall VRE also declines a bit, leading to longer use of natural gas and a bit slower electrification.

Thus, even in this unlikely scenario where integration challenges are twice the default, solar and wind remain highly competitive, with limited implications to the overall mitigation strategy. The delayed reduction in natural gas use results in an increase in emissions of 1–2% by 2040, shifting the lower bound of the 2040 emissions reduction range from 82% to 80% (Figure 1).

Figure 1. 2040 emission reductions per scenario

Also, as shown in Figure 2, variable renewables stay mostly below 80% of electricity generation by 2040, and around 85% by 2050, at which point additional flexibility measures, such as green hydrogen, are expected to be available in higher volumes. For context, Denmark already achieved roughly 70% wind and solar in its electricity mix in 2024, although clearly for Denmark the integration challenge of having high VRE shares is smaller due to its strong grid connections to (larger) neighbouring countries with currently still lower VRE shares that can act as buffers.

Figure 2. Wind and solar share in electricity generation.

We are continuously working on validating and improving the representation of wind & solar integration in REMIND, either by validating the parameterization against the results of detailed power sector models, or by actually coupling REMIND to a detailed power sector model. As you correctly mention, we have achieved this for a one-region version for Germany, and found that our REMIND-internal dynamics resulted in quite similar aggregated results, such as overall generation shares and costs, when compared to the detailed coupled version. We are still working on developing a full coupling for further regions, so we sadly cannot represent detailed hourly modelling results in the current study.

We have now added a paragraph in the supplementary material describing the representation of VRE integration in REMIND.

Regarding the description on Efficiency case in Table 1, would it be possible to provide final energy amount in Exajoule (or Petajoule) for comparison with other figures or tables in this manuscript which are presented in EJ.

EJ values were added to table 1.

Reviewer #5

Remarks to the Author:

Thank you for the revisions to this timely and important manuscript. It comes at a critical juncture in EU climate policy development, as policy and civil society debate on the 2040 greenhouse gas reduction target proposed by the European Commission is in full swing, yet robust quantitative analyses on the proposal - to contrast with the Commission's own impact assessment - are scarce. Hopefully the mainly results have also been fed into the formal consultation process.

I have reviewed the feedback by the previous reviewers, some of which was appropriate, some of which seemed rather extraneous, digressive or tangential. The authors' responses and manuscript modifications are comprehensive, in almost all cases addressing the reviewers' suggestions or criticisms, or in rare cases justifying the decision not to follow the reviewer recommendation (but even then often adding language to clarify a point, or highlight issues not covered by the present manuscript).

I do not see any potential to further improve the manuscript without tradeoffs - e.g. deepening analysis of one aspect and in the process sacrificing focus and depth elsewhere. I also believe further time invested chasing marginal improvements will be outweighed by the missed opportunity to inform a very important policy process with a directly relevant, peer reviewed quantitative analysis. Hence my recommendation is to accept 'as is'.

We thank the reviewer for the positive review.